# Comprehensive machine learning analysis of *Hydra* behavior reveals a stable basal behavioral repertoire

**Shuting Han\*, Ekaterina Taralova, Christophe Dupre, Rafael Yuste**

NeuroTechnology Center, Department of Biological Sciences, Columbia University, New York, United States

**Abstract** Animal behavior has been studied for centuries, but few efficient methods are available to automatically identify and classify it. Quantitative behavioral studies have been hindered by the subjective and imprecise nature of human observation, and the slow speed of annotating behavioral data. Here, we developed an automatic behavior analysis pipeline for the cnidarian *Hydra vulgaris* using machine learning. We imaged freely behaving *Hydra*, extracted motion and shape features from the videos, and constructed a dictionary of visual features to classify pre-defined behaviors. We also identified unannotated behaviors with unsupervised methods. Using this analysis pipeline, we quantified 6 basic behaviors and found surprisingly similar behavior statistics across animals within the same species, regardless of experimental conditions. Our analysis indicates that the fundamental behavioral repertoire of *Hydra* is stable. This robustness could reflect a homeostatic neural control of "housekeeping" behaviors which could have been already present in the earliest nervous systems.

DOI: https://doi.org/10.7554/eLife.32605.001

## Introduction

Animal behavior is generally characterized by an enormous variability in posture and the motion of different body parts, even if many complex behaviors can be reduced to sequences of simple stereotypical movements (*Berman et al., 2014*; *Branson et al., 2009*; *Gallagher et al., 2013*; *Srivastava et al., 2009*; *Wiltschko et al., 2015*; *Yamamoto and Koganezawa, 2013*). As a way to systematic capture this variability and compositionality, quantitative behavior recognition and measurement methods could provide an important tool for investigating behavioral differences under various conditions using large datasets, allowing for the discovery of behavior features that are beyond the capability of human inspection, and defining a uniform standard for describing behaviors across conditions (*Egnor and Branson, 2016*). In addition, much remains unknown about how the specific spatiotemporal pattern of activity of the nervous systems integrate external sensory inputs and internal neural network states in order to selectively generate different behavior. Thus, automatic methods to measure and classify behavior quantitatively could allow researchers to indetify potential neural mechanisms by providing a standard measurement of the behavioral output of the nervous system.

Indeed, advances in calcium imaging techniques have enabled the recording of the activity of large neural populations (*Chen et al., 2013*; *Jin et al., 2012*; *Kralj et al., 2011*; *St-Pierre et al., 2014*; *Tian et al., 2009*; *Yuste and Katz, 1991*), including whole brain activity from small organisms such as *C. elegans* and larval zebrafish (*Ahrens et al., 2013*; *Nguyen et al., 2016*; *Prevedel et al., 2014*). A recent study has demonstrated the cnidarian *Hydra* can be used as an alternative model to image the complete neural activity during behavior (*Dupre and Yuste, 2017*). As a cnidarian, *Hydra* is close to the earliest animals in evolution that had nervous systems. As the output of the nervous

**\*For correspondence:** shuting.han@columbia.edu

**Competing interests:** The authors declare that no competing interests exist.

**eLife digest** How do animals control their behavior? Scientists have been trying to answer this question for over 2,000 years, and many studies have analysed specific behaviors in different animals. However, most of these studies have traditionally relied on human observers to recognise and classify different behaviors such as movement, rest, grooming or feeding. This approach is subject to human error and bias, and is also very time consuming. Because of this, reseachers normally only study one particular behavior, in a piecemeal fashion. But to capture all the different actions an animal generates, faster, more objective methods of systematically classifying and quantifying behavior would be ideal.

One promising opportunity comes from studying a small freshwater organism called *Hydra*, one of the most primitive animals with a nervous system. Thanks to *Hydra*'s transparent body, modern imaging techniques can be used to observe the activity of their whole nervous system all at once, while the animal is engaged in different actions. However, to realise this potential, scientists need a quick way of automatically recognising different *Hydra* behaviors, such as contracting, bending, tentacle swaying, feeding or somersaulting. This is particularly difficult because *Hydra*'s bodies can change shape in different situations.

To address this, Han et al. borrowed cutting-edge techniques from the field of computer vision to create a computer program that could automatically analyse hours of videos of freely-moving *Hydra* and classify their behavior automatically. The computer algorithms can learn how to recognise different behaviors in two ways: by learning from examples already classified by humans (known as 'supervised learning') or by letting it pick out different patterns by itself (known as 'unsupervised learning'). The program was able to identify all the behaviors previously classified by humans, as well as new types that had been missed by human observation.

Using this new computer program, Han et al. discovered that *Hydra*'s collection of six basic behaviors stays essentially the same under different environmental conditions, such as light or darkness. One possible explanation for this is that its nervous system adapts to the environment to maintain a basic set of actions it needs for survival, although another possibility is that *Hydra* just does not care and goes along with its basic behaviors, regardless of the environment. Han et al.'s new method is useful not only for classifying all behavioral responses in *Hydra*, but could potentially be adapted to study all the behaviors in other animal species. This would allow scientists to systematically perform experiments to understand how the nervous system controls all animal behavior, a goal that it is the holy grail of neuroscience.

DOI: https://doi.org/10.7554/eLife.32605.002

system, animal behavior allows individuals to adapt to the environment at a time scale that is much faster than natural selection, and drives the rapid evolution of the nervous system, providing a rich context to study nervous system functions and evolution (*Anderson and Perona, 2014*). As *Hydra*'s nervous system evolved from that present in the last common ancestor of cnidarians and bilaterians, the behaviors of *Hydra* could also represent some of the most primitive examples of coordination between a nervous system and non-neuronal cells. This could make *Hydra* particularly relevant to our understanding of the nervous systems of model organisms such as *Caenorhabditis elegans*, *Drosophila*, zebrafish, and mice, as it provides an evolutionary perspective to discern whether neural mechanisms found in those species represent a specialization or are generally conserved. In fact, although *Hydra* behavior has been study for centuries, it is still unknown whether *Hydra* possesses complex behaviors such as social interactions and learning, how its behavior changes under environmental, physiological, nutritional or pharmacological manipulations, or what are the underlying neural mechanisms of these potential changes. Having an unbiased and automated behavior recognition and quantification method would therefore enable such studies with large datasets. This could allow high-throughput systematic pharmacological assays, lesion studies, environmental and physiological condition changes in behavior, or alternations under activation of subsets of neurons, testing quantitative models, and linking behavior outputs with the underlying neural activity patterns.

*Hydra* behavior was first described by *Trembley (1744)*, and it consists of both spontaneous and stimulus-evoked movements. Spontaneous behaviors include contraction (*Passano and McCullough,*

1964) and locomotion such as somersaulting and inchworming (*Mackie, 1974*), and can sometimes be induced by mechanical stimuli or light. Food-associated stimuli induce a stereotypical feeding response that consists of three distinct stages: tentacle writhing, tentacle ball formation and mouth opening (*Koizumi et al., 1983*; *Lenhoff, 1968*). This elaborate reflex-like behavior is fundamental to the survival of *Hydra* and sensitive to its needs: well-fed animals do not appear to show feeding behavior when exposed to a food stimulus (*Lenhoff and Loomis, 1961*). In addition, feeding behavior can be robustly induced by small molecules such as glutathione and S-methyl-glutathione (GSM) (*Lenhoff and Lenhoff, 1986*). Besides these relatively complex behaviors, *Hydra* also exhibits simpler behaviors with different amplitudes and in different body regions, such as bending, individual tentacle movement, and radial and longitudinal contractions. These simpler behaviors can be oscillatory and occur in an overlapping fashion and are often hard to describe in a quantitative manner. This, in turn, makes complex behaviors such as social or learning behaviors, which can be considered as sequences of simple behaviors, hard to quantitatively define. Indeed, to manually annotate behaviors in videos that are hours or days long is not only extremely time-consuming, but also partly subjective and imprecise (*Anderson and Perona, 2014*). However, analyzing large datasets of behaviors is necessary to systematically study behaviors across individuals in a long-term fashion. Recently, computational methods have been developed to define and recognize some behaviors of *C. elegans* (*Brown et al., 2013*; *Stephens et al., 2008*) and *Drosophila* (*Berman et al., 2014*; *Johnson et al., 2016*). These pioneer studies identify the movements of animals by generating a series of posture templates and decomposing the animal posture at each time points with these standard templates. This general framework works well for animals with relatively fixed shapes. However, *Hydra* has a highly deformable body shape that contracts, bends and elongates in a continuous and non-isometric manner, and the same behavior can occur at various body postures. Moreover, Hydra has different numbers of tentacles and buds across individuals, which presents further challenges for applying template-based methods. Therefore, a method that encodes behavior information in a statistical rather than an explicit manner is desirable.

As a potential solution to this challenge, the field of computer vision has recently developed algorithms for deformable human body recognition and action classification. Human actions have large variations based on the individual's appearance, speed, the strength of the action, background, illumination, etc. (*Wang et al., 2011*). To recognize the same action across conditions, features from different videos need to be represented in a unified way. In particular, the Bag-of-Words model (BoW model) (*Matikainen et al., 2009*; *Sun et al., 2009*; *Venegas-Barrera and Manjarrez, 2011*; *Wang et al., 2011*) has become a standard method for computer vision, as it is a video representation approach that captures the general statistics of image features in videos by treating videos as 'bags' of those features. This enables to generalize behavior features in a dataset that is rich with widely varied individual-specific characteristics. The BoW model originated from document classification and spam-detection algorithms, where a text is represented by an empirical distribution of its words. To analyze videos of moving scenes, the BoW model has two steps: feature representation and codebook representation. In the first step, features (i.e. 'words' such as movements and shapes) are extracted and unified into descriptor representations. In the second step, these higher order descriptors from multiple samples are clustered (i.e. movement motifs), usually by k-means algorithms, and then averaged descriptors from each cluster are defined as 'codewords' that form a large codebook. This codebook in principle contains representative descriptors of all the different movements of the animal. Therefore, each clip of the video can be represented as a histogram over all codewords in the codebook. These histogram representations can be then used to train classifiers such as SVMs, or as inputs to various clustering algorithms, supervised or unsupervised, to identify and quantify behavior types. BoW produces an abstract representation compared to manually specified features, and effectively leverages the salient statistics of the data, enabling modeling of large populations. Doing so on a large scale with manually selected features is not practical. The power of such a generalization makes the BoW framework particularly well suited for addressing the challenge of quantifying *Hydra* behavior.

Inspired by previous work on *C. elegans* (*Brown et al., 2013*; *Kato et al., 2015*; *Stephens et al., 2008*) and *Drosophila* (*Berman et al., 2014*; *Johnson et al., 2016*; *Robie et al., 2017*) as well as by progress in computer vision (*Wang et al., 2011*), we explored the BoW approach, combining computer vision and machine learning techniques, to identify both known and unannotated behavior types in *Hydra*. To do so, we imaged behaviors from freely moving *Hydra*, extracted motion and

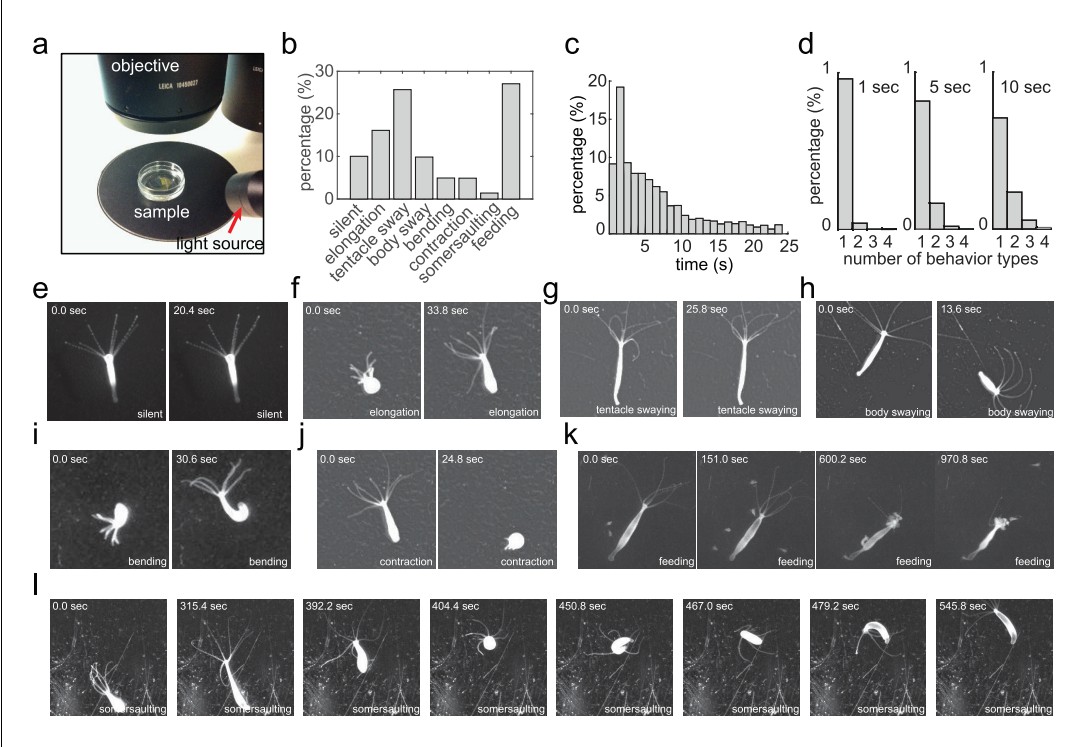

**Figure 1.** Acquiring an annotated *Hydra* behavior dataset. (**a**) Imaging *Hydra* behavior with a widefield dissecting microscope. A *Hydra* polyp was allowed to move freely in a Petri dish, which was placed on a dark surface under the microscope objective. The light source was placed laterally, creating an bright image of the *Hydra* polyp on a dark background. (**b**) Histogram of the eight annotated behavior types in all data sets. (**c**) Histogram of the duration of annotated behaviors. (**d**) Histogram of total number of different behavior types in 1 s, 5 s and 10 s time windows. (**e–l**) Representative images of silent (**e**), elongation (**f**), tentacle swaying (**g**), body swaying (**h**), bending (**i**), contraction (**j**), feeding (**k**), and somersaulting (**l**) behaviors.
DOI: https://doi.org/10.7554/eLife.32605.003

The following figure supplement is available for figure 1:

**Figure supplement 1.** Variability of human annotators.
DOI: https://doi.org/10.7554/eLife.32605.004

shape features from the videos, and constructed a dictionary of these features. We then trained classifiers to recognize *Hydra* behavior types with manual annotations, and identified both annotated and unannotated behavior types in the embedding space. We confirmed the performance of the algorithms with manually annotated data and then used the method for a comprehensive survey of *Hydra* behavior, finding a surprising stability in the expression of six basic behaviors, regardless of the different experimental and environmental conditions. These findings are consistent with the robust behavioral and neural circuit homeostasis found in other invertebrate nervous systems for "housekeeping" functions (*Haddad and Marder, 2017*).

## Results

### Capturing the movement and shape statistics of freely moving *Hydra*

Our goal was to develop a method to characterize the complete behavioral repertoire of *Hydra* under different laboratory conditions. We collected a *Hydra* behavior video dataset (*Han, 2018a*) using a widefield dissecting microscope, allowing *Hydra* to move freely in a culture dish (*Figure 1a*). We imaged 53 *Hydra* specimens at a rate of 5 Hz for 30 min, and we either allowed each of them to behave freely, or induced feeding behavior with glutathione, since feeding could not be observed without the presence of prey (which would have obscured the imaging). From viewing these data, we visually identified eight different behaviors, and manually annotated every frame of the entire dataset with the following labels for these eight behavioral states: silent (no apparent motion),

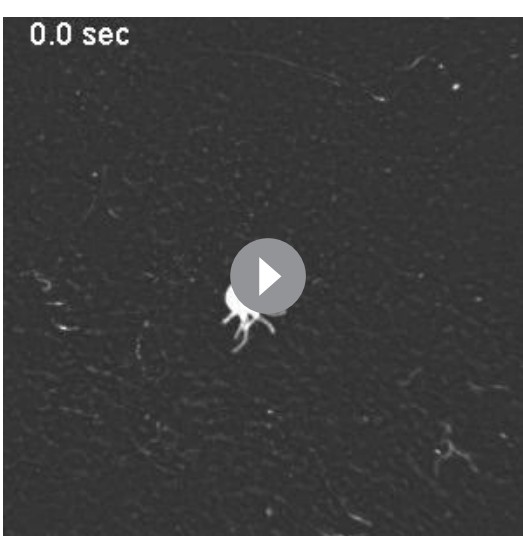

**Video 1.** Example of elongation behavior. The animal was allowed to move freely in a petri dish. The video was taken at 5 Hz, and was accelerated 20 fold.
DOI: https://doi.org/10.7554/eLife.32605.005

elongation, tentacle swaying, body swaying, bending, contraction, somersaulting, and feeding (*Figure 1b*; *Figure 1e-l*; *Videos 1–7*). Overall, we acquired an annotated *Hydra* behavior dataset with 360,000 fames in total. We noticed that most behaviors in our manual annotation lasted less than 10 s (*Figure 1c*), and that, within a time window of 5 s, most windows contained only one type of behavior (*Figure 1d*). A post-hoc comparison of different window sizes (1–20 s) with the complete analysis framework also demonstrated that 5 s windows result in the best performance (*Figure 2—figure supplement 1a*). Therefore, we chose 5 s as the analysis length of a behavior element in *Hydra*.

Due to the large shape variability of the highly deformable *Hydra* body during behavior, methods that construct postural eigenmodes from animal postures are not suitable. Therefore, we designed a novel pipeline consisting of four steps: pre-processing, feature extraction, codebook generation, and feature encoding (*Han, 2018b*) (*Figure 2*), in line with the BoW framework. Pre-processing was done to exclude the variability in size and rotation angle during imaging, which introduces large variance. To do so, we first defined a behavior element as a 5 s time window, splitting each behavior video into windows accordingly. Then we fitted the body column of *Hydra* into an ellipse, and centered, rotated, and scaled the ellipse to a uniform template ellipse in each element window. We then encoded spatial information into the BoW framework by segmenting the *Hydra* area through the videos with an automated program, dividing it into a tentacle region, an upper body region, and a lower body region (Materials and methods; *Video 8*).

After this encoding, in a feature extraction step we applied a dense trajectory method in each 5 s window element (*Wang et al., 2011*). This dense trajectory method represents video patches by several shape and motion descriptors, including a Histogram of Oriented Gradient (HOG) (*Dalal and Triggs, 2005*), which is based on edge properties in the image patch; and a Histogram of Optical Flow (HOF) as well as a Motion Boundary Histogram (MBH) (*Dalal et al., 2006*), based on motion properties. With the dense trajectory method, we first detected and tracked points with prominent features throughout the videos. Then, for each feature point, we analyzed a small surrounding local patch and computed the motion and shape information therein represented by HOF, HOG and MBH descriptors (*Video 9*). Thus, each video window element was captured as motion and shape descriptors associated with a set of local video patches with distinguished visual features.

To quantize the 'bags' of features from each element time window, we collected a uniform feature codebook using all the dense trajectory features. Intuitively, the elements in the codebook are the representative features for each type of motion or shape in a local patch, therefore they can be regarded as standard entries in a dictionary. Here, we generate the codebook in a 'soft' manner, where the codebook contains information of the centroid of clusters and their shape. We fitted the features with k Gaussian mixtures. Because each Gaussian is characterized not only by its mean, but also by its variance, we preserved more information than with other 'hard' methods like k-means. The next step was to encode the features with the codebook. For this, 'hard' methods where one encodes the features by assigning each feature vector to its nearest Gaussian mixture, lose information concerning the shapes of the Gaussians. To avoid this, we encoded the features using Fisher vectors, which describe the distance between features and the Gaussian mixture codebook entries in a probabilistic way, encoding both the number of occurrence and the distribution of the descriptors (*Perronnin et al., 2010*) (*Figure 2—figure supplement 1b*). Since each element window was split into tentacle, upper body and lower body region, we were able to integrate spatial information by encoding the features in each of the three body regions separately (*Figure 2—figure*

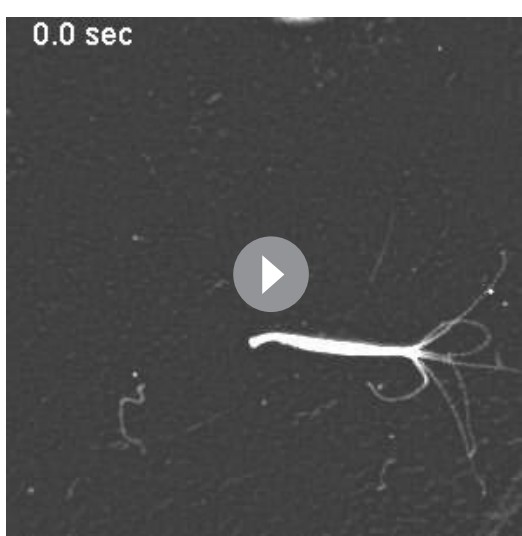

**Video 2.** Example of tentacle swaying behavior. The animal was allowed to move freely in a petri dish. The video was taken at 5 Hz, and was accelerated 20 fold. DOI: https://doi.org/10.7554/eLife.32605.006

supplement 1b). Finally, we represented the behavior in each element window by the concatenated Fisher vector from the three regions.

## *Hydra* behavior classified from video statistics

Like all animals, *Hydra* exhibits behaviors at various time scales. Basic behaviors such as elongation and bending are usually long and temporally uniform, while tentacle swaying, body swaying and contraction are usually short and executed in a burst-like manner. Feeding and somersaulting are more complex behaviors that can be broken down into short behavior motifs (*Videos 6–7*) (*Lenhoff and Loomis, 1961*). Feeding is apparently a stepwise, fixed action pattern-like uniform behavior, with smooth transitions between tentacle writhing, ball formation, and mouth opening (*Video 6*). Somersaulting represents another fixed action pattern-like behavior and typically consists of a sequence of basic behaviors with elongation accompanied by tentacle movements, contraction, bending, contraction, elongation, and contraction; completing the entire sequence takes a few minutes (*Video 7*). The time spent during each step and the exact way each step is executed vary between animals. Thus, to study *Hydra* behavior, it is essential to accurately recognize the basic behavior types that comprise these complex activities.

We aimed to capture basic behaviors including silent, elongation, tentacle swaying, body swaying, bending, contraction, and feeding, using the Fisher vector features that encode the video statistics. These features were extracted from 5 s element windows and exhibited stronger similarity within the same behavior type, but were

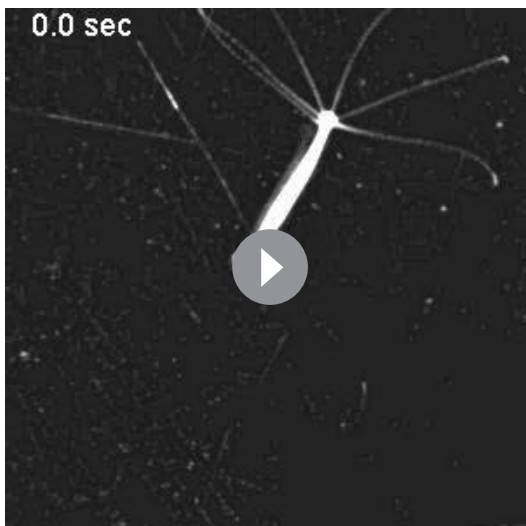

**Video 3.** Example of body swaying behavior. The animal was allowed to move freely in a petri dish. The video was taken at 5 Hz, and was accelerated 20 fold. DOI: https://doi.org/10.7554/eLife.32605.007

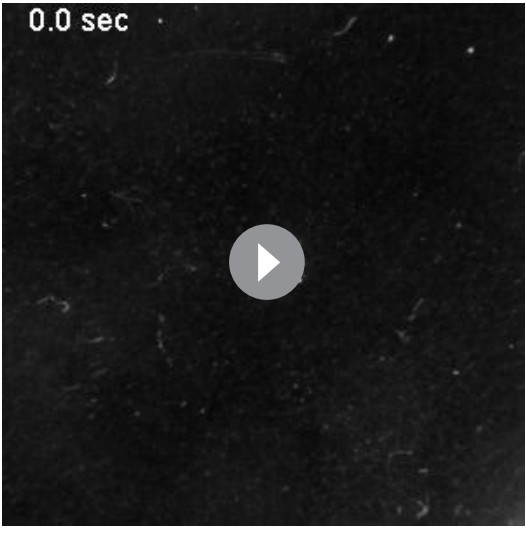

**Video 4.** Example of bending behavior. The animal was allowed to move freely in a petri dish. The video was taken at 5 Hz, and was accelerated 20 fold. DOI: https://doi.org/10.7554/eLife.32605.008

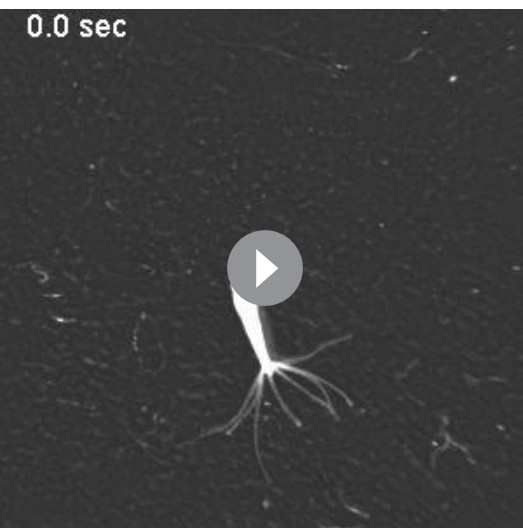

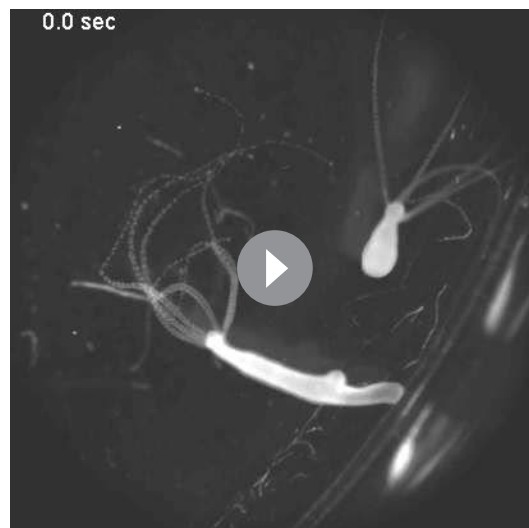

**Video 5.** Example of a contraction burst. The animal was allowed to move freely in a petri dish. The video was taken at 5 Hz, and was accelerated 20 fold.
DOI: https://doi.org/10.7554/eLife.32605.009

**Video 6.** Example of induced feeding behavior. The animal was treated with reduced L-glutathione at 45 s. The video was taken at 5 Hz, and was accelerated 20 fold.
DOI: https://doi.org/10.7554/eLife.32605.010

distinguished from features of different behavior types (*Figure 3a*). We then trained support vector machine (SVM) classifiers with manual labels on data from 50 *Hydra*, and tested them on a random 10% withheld validation dataset. We evaluated classification performance via the standard receiver operating characteristic (ROC) curve and area under curve (AUC). In addition, we calculated three standard measurements from the number of true positives (TP), true negatives (TN), false positives (FP), and false negatives (FN): accuracy, defined as (TP + TN)/(TP + TN + FP + FN); precision, defined as TP/(TP + FP); and recall, defined as TP/(TP + FN). We achieved perfect training performance (AUC = 1, accuracy 100%), while on the validation data the overall accuracy was 86.8%, and mean AUC was 0.97 (*Figure 3b and c*; *Table 1*).

This classification framework was easily generalized to new data. With data from three *Hydra* that were not involved in either codebook generation or classifier training, we extracted and encoded features using the generated codebook, and achieved classification accuracy of 90.3% for silent (AUC = 0.95), 87.9% for elongation (AUC = 0.91), 71.9% for tentacle swaying (AUC = 0.76), 83.4% for body swaying (AUC = 0.75), 93.9% for bending (AUC = 0.81) and 92.8% for contraction (AUC = 0.92). All the classifiers achieved significantly better performance than chance levels (*Figure 3b, c and d*; *Table 1*; *Video 10*). Interestingly, the variability in classifier performance with new data matched human annotator variability (*Figure 1—figure supplement 1*). This demonstrates that the codebook generated from training data efficiently captured *Hydra* behaviors and that trained classifiers can robustly identify the basic behaviors of *Hydra* and predict their occurrence automatically from the data.

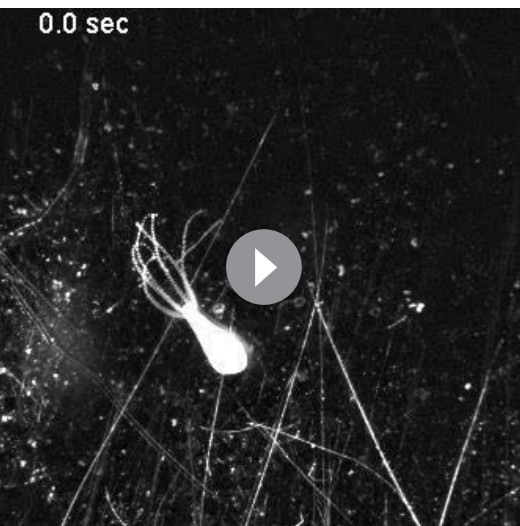

**Video 7.** Example of somersaulting behavior. The video was taken at 5 Hz, and was accelerated by 20 fold.
DOI: https://doi.org/10.7554/eLife.32605.011

*Hydra* can exhibit overlapping behaviors at the same time. For example, a *Hydra* specimen could be moving its tentacles while bending, or swaying its body while elongating. In such cases, it would be imprecise to allow only a single behavior label per time window. To capture this situation, we allowed a 'soft' classification strategy, taking up to three highest classification types that have a classifier probability within a twofold difference between them. With joint classifiers, we achieved 86.8% overall accuracy on the validation data (81.6% with hard classification), and 59.0% with new test data (50.1% with hard classification). Soft classification improved classification performance by allowing a realistic situation when *Hydra* transitions between two behaviors, or executes multiple behaviors simultaneously.

In addition to optimally classifying the seven basic behaviors described above, classifying somersaulting video clips with basic behavior classifiers showed a conserved structure during the progression of this behavior (*Figure 3e*; *Video 11*). Somersaulting is a complex behavioral sequence that was not included in the seven visually identified behavior types. This long behavior can typically be decomposed into a sequence of simple behaviors of tentacle swaying, elongation, body swaying, contraction, and elongation. Indeed, in our classification of somersaulting with the seven basic behavior types, we noticed a strong corresponding structure: the classified sequences start with tentacle swaying, elongation, and body swaying, then a sequence of contraction and elongation before a core bending event (*Figure 3e*); finally, elongation and contraction complete the entire somersaulting behavior. This segmented classification based on breaking down a complex behavior into a sequence of multiple elementary behaviors agrees with human observations, indicating that our method is able to describe combined behaviors using the language of basic behavior types.

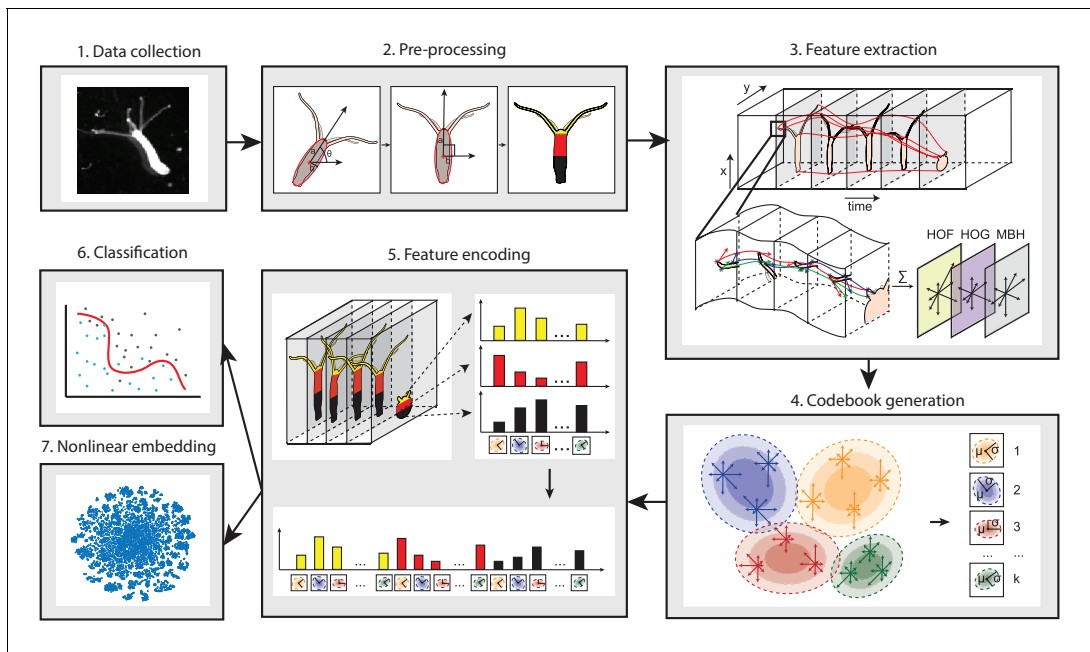

**Figure 2.** Analysis pipeline. Videos of freely moving *Hydra* polyps were collected (1), then, *Hydra* images were segmented from background, and the body column was fit to an ellipse. Each time window was then centered and registered, and the *Hydra* region was separated into three separate body parts: tentacles, upper body column, and lower body column (2). Interest points were then detected and tracked through each time window, and HOF, HOG and MBH features were extracted from local video patches of interest points. Gaussian mixture codebooks were then generated for each features subtype (4), and Fisher vectors were calculated using the codebooks (5). Supervised learning using SVM (6), or unsupervised learning using t-SNE embedding (7) was performed using Fisher vector representations.
DOI: https://doi.org/10.7554/eLife.32605.012
The following figure supplement is available for figure 2:

**Figure supplement 1.** Model and parameter selection.
DOI: https://doi.org/10.7554/eLife.32605.013

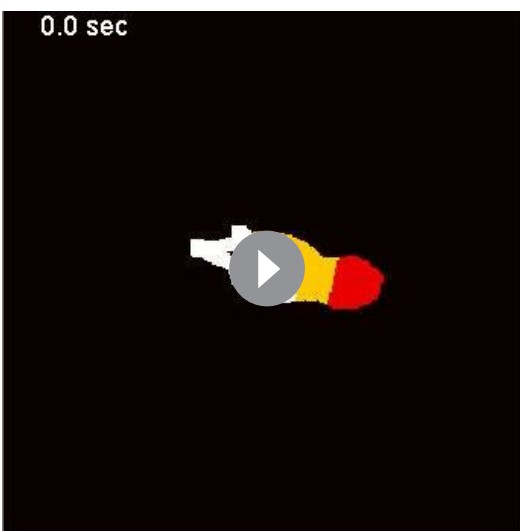

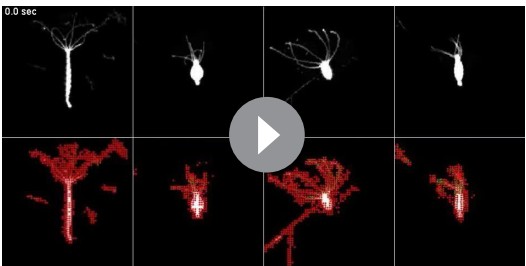

**Video 9.** Examples of detected interest points (red) and dense trajectories (green) in tentacle swaying (left), elongation (middle left), body swaying (middle right), and contraction (right) behaviors in 2 s video clips. Upper panels show the original videos; lower panels show the detected features.
DOI: https://doi.org/10.7554/eLife.32605.015

**Video 8.** Example of the output of body part segmentation. White represents tentacle region, yellow represents upper body column region, and red represents lower body column region.
DOI: https://doi.org/10.7554/eLife.32605.014

## Unsupervised discovery of behavior states in embedding space

Manual annotation identifies behavior types on the basis of distinct visual features. However, it is subjective by nature, especially when the *Hydra* exhibits multiple behaviors simultaneously and can be affected by the individual biases of the annotator. Therefore, to complement the supervised method described above, where classifiers were trained with annotated categories, we sought to perform unsupervised learning to discover the structural features of *Hydra* behaviors. Since the Fisher vector representation of video statistics is high-dimensional, we applied a nonlinear embedding technique, t-Distributed Stochastic Neighbor Embedding (t-SNE), to reduce the feature vector dimensionality (*Berman et al., 2014*; *Van Der Maaten, 2009*). This also allowed us to directly visualize the data structure in two dimensions while preserving the local structures in the data, serving as a method for revealing potential structures of the behavior dataset.

Embedding the feature vectors of training data resulted in a t-SNE map that corresponded well to our manual annotation (*Figure 4a*). Generating a density map over the embedded data points revealed cluster-like structures in the embedding space (*Figure 4b*). We segmented the density map into regions with a watershed method, which defined each region as a behavior motif region (*Figure 4c and e*). We evaluated the embedding results by quantifying the manual labels of data points in each behavior motif region. We then assigned a label to each region based on the majority of the manually labeled behavior types in it. Using this approach, we identified 10 distinct behavior regions in the map (*Figure 4d*). These regions represented not only the seven types we defined for supervised learning, but also a somersaulting region, and three separate regions representing the three stages of feeding behavior (*Figure 4d*). Embedding with continuous 5 s time windows, which exclude the effect of the hard boundaries of separating the behavior elements, revealed the same types of behaviors (*Figure 4—figure supplement 1*).

The generated embedding space could be used to embed new data points (*Berman et al., 2014*). We embedded feature vectors from a withheld validation dataset, as well as from three *Hydra* that were involved neither in generating the feature codebook, nor in the embedding space generation (*Figure 4f*). Quantitative evaluation of embedding performance with manual labels showed that all behavior types were accurately identified by embedding in the validation data. In test samples, embedding identification of elongation, tentacle sway, body sway, contraction, and the ball formation stage of feeding, all agreed with manual labels (*Figure 4g*). Therefore, embedding of feature vectors can identify the same behavior types that are identified by human annotation.

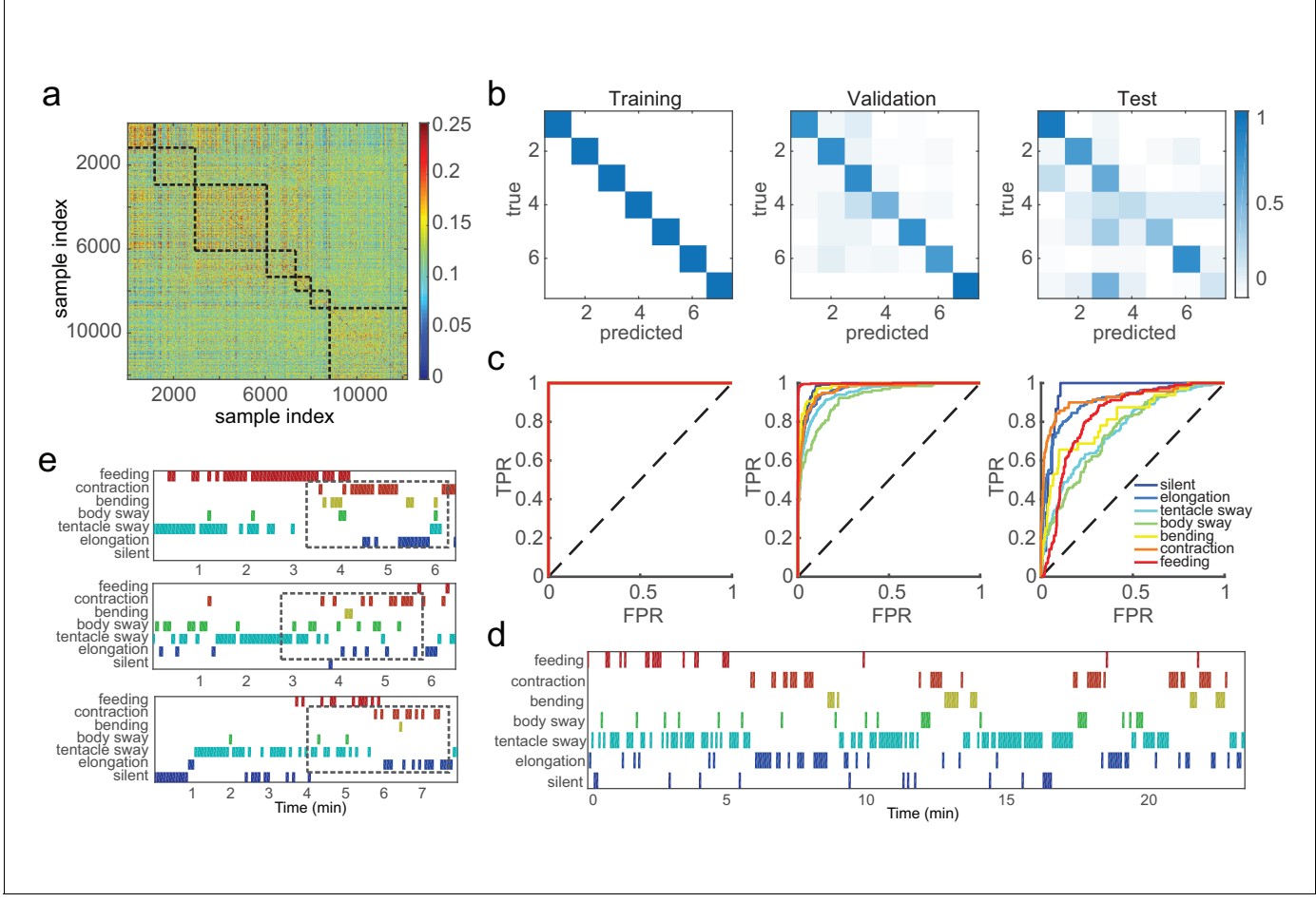

**Figure 3.** SVM classifiers recognize pre-defined *Hydra* behavior types. (**a**) Pairwise Euclidean similarity matrix of extracted Fisher vectors. Similarity values are indicated by color code. (**b**) Confusion matrices of trained classifiers predicting training, validation, and test data. Each column of the matrix represents the number in a predicted class; each row represents the number in a true class. Numbers are color coded as color bar indicates. (Training: n = 50, randomly selected 90% samples; validation: n = 50, randomly selected 10% samples; test: n = 3) (**c**) ROC curves of trained classifiers predicting training, validation and test data. TPR, true positive rate; FPR, false positive rate. Dashed lines represent chance level. (**d**) An example of predicted ethogram using the trained classifiers. (**e**) Three examples of SVM classification of somersaulting behaviors. Dashed boxes indicate the core bending and flipping events.

DOI: https://doi.org/10.7554/eLife.32605.016

**Table 1.** SVM statistics. AUC: area under curve; Acc: accuracy; Prc: precision; Rec: recall.

| | Train | | | | | | Withheld | | | | | | Test | | | | | |
|---|---|---|---|---|---|---|---|---|---|---|---|---|---|---|---|---|---|---|
| Behavior | AUC | AUC chance | Acc | Acc chance | Prc | Rec | AUC | AUC chance | Acc | Acc chance | Prc | Rec | AUC | AUC chance | Acc | Acc chance | Prc | Rec |
| Silent | 1 | 0.5 | 100% | 9.6% | 100% | 100% | 0.98 | 0.5 | 95.6% | 9.6% | 75.6% | 97.4% | 0.95 | 0.5 | 90.3% | 1.9% | 18.4% | 90.3% |
| Elongation | 1 | 0.5 | 100% | 14.2% | 100% | 100% | 0.96 | 0.5 | 93.4% | 13.6% | 76.4% | 95.9% | 0.91 | 0.5 | 87.9% | 22.2% | 71.4% | 92.6% |
| Tentacle sway | 1 | 0.5 | 100% | 25.1% | 100% | 100% | 0.95 | 0.5 | 89.6% | 25.0% | 77.5% | 92.4% | 0.76 | 0.5 | 71.9% | 30.2% | 47.9% | 76.7% |
| Body sway | 1 | 0.5 | 100% | 10.0% | 100% | 100% | 0.92 | 0.5 | 92.9% | 9.3% | 65.7% | 97.0% | 0.75 | 0.5 | 83.4% | 17.7% | 52.8% | 95.4% |
| Bending | 1 | 0.5 | 100% | 5.2% | 100% | 100% | 0.98 | 0.5 | 97.3% | 6.1% | 74.4% | 98.4% | 0.81 | 0.5 | 93.9% | 6.1% | 38.9% | 96.5% |
| Contraction | 1 | 0.5 | 100% | 6.6% | 100% | 100% | 0.97 | 0.5 | 95.7% | 6.9% | 70.4% | 97.7% | 0.92 | 0.5 | 92.8% | 11.7% | 63.2% | 95.5% |
| Feeding | 1 | 0.5 | 100% | 29.2% | 100% | 100% | 1 | 0.5 | 98.8% | 29.6% | 98.5% | 99.4% | 0.83 | 0.5 | 81.0% | 10.2% | 39.6% | 94.1% |

DOI: https://doi.org/10.7554/eLife.32605.017

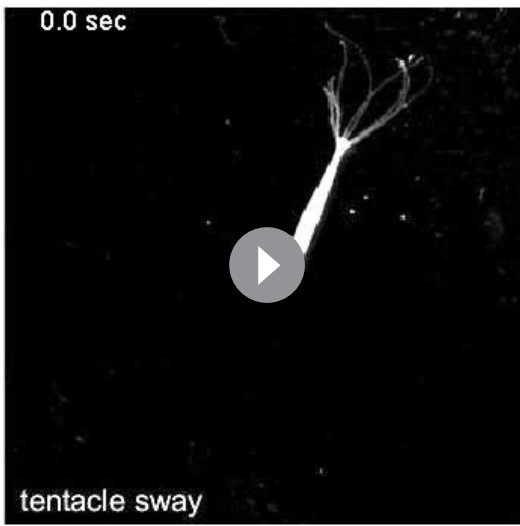

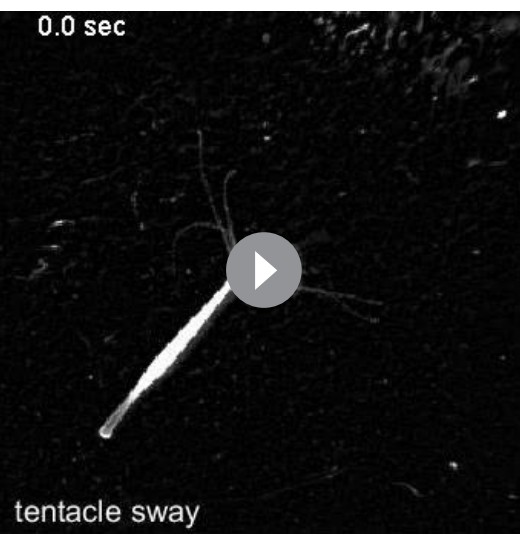

**Video 10.** Example of the trained SVM classifiers predicting new data.
DOI: https://doi.org/10.7554/eLife.32605.018

**Video 11.** Example of the trained SVM classifiers predicting somersaulting behavior from a new video. Soft prediction was allowed here.
DOI: https://doi.org/10.7554/eLife.32605.019

## Embedding reveals unannotated behaviors in long datasets

We wondered if *Hydra* has any spontaneous behaviors under natural day/night cycles that were not included in our manually labeled sets. We mimicked natural conditions by imaging a *Hydra* polyp for 3 days and nights with a 12 hr dark/light cycle (*Figure 5a*), keeping the *Hydra* in a 100 μm thick coverslip covered chamber to constrain it within the field of view of the microscope (*Figure 5b*) (*Dupre and Yuste, 2017*). This imaging approach, although constraining the movement of *Hydra*, efficiently reduced the complexity of the resulting motion from a three-dimensional to a two-dimensional projection, while still allowing the *Hydra* to exhibit a basic repertoire of normal behaviors.

Using this new dataset, we generated a t-SNE embedding density map from the feature vectors as previously described, and segmented it into behavior motif regions (*Figure 5c*). Among the resulting 260 motif regions, we not only discovered previously defined behavior types including silent, elongation, bending, tentacle swaying, and contraction, but also found subtypes within certain classes (*Videos 12–19*). In elongation, for example, we found three different subtypes based on the state of the animal: slow elongation during the resting state of the animal, fast elongation after a contraction burst, and inter-contraction elongation during a contraction burst (*Videos 13–15*). In contraction, we found two different subtypes: the initial contraction of a contraction burst, and the subsequent individual contraction events when the animal is in a contracted state (*Videos 18–19*). Interestingly, we also discovered one region in the embedding map that showed a previously unannotated egestion behavior (*Figure 5c*; *Video 20*). Egestion behavior (also known as radial contraction) has been observed before (*Dupre and Yuste, 2017*), and is typically a fast, radial contraction of the body column that happens within 1 s and empties the body cavity of fluid. Although this behavior happens with animals in their natural free movement, its fast time scale and the unconstrained movement make it hard to identify visually during human annotation. In addition, another t-SNE region showed a novel hypostome movement associated with egestion, characterized by a regional pumping-like movement in hypostome and lower tentacle regions (*Video 21*).

We evaluated the reliability of the identification of this newly discovered egestion behavior from the embedding method by detecting egestion with an additional ad-hoc method. We measured the width of the *Hydra* body column by fitting it to an ellipse, and low-pass filtered the width trace. Peaks in the trace then represent estimated time points of egestion behavior, which is essentially a rapid decrease in the body column width (*Figure 5d*). Detected egestion time points were densely distributed in the newly discovered egestion region in the embedding map (*Figure 5e*), confirming that our method is as an efficient way to find novel behavior types.

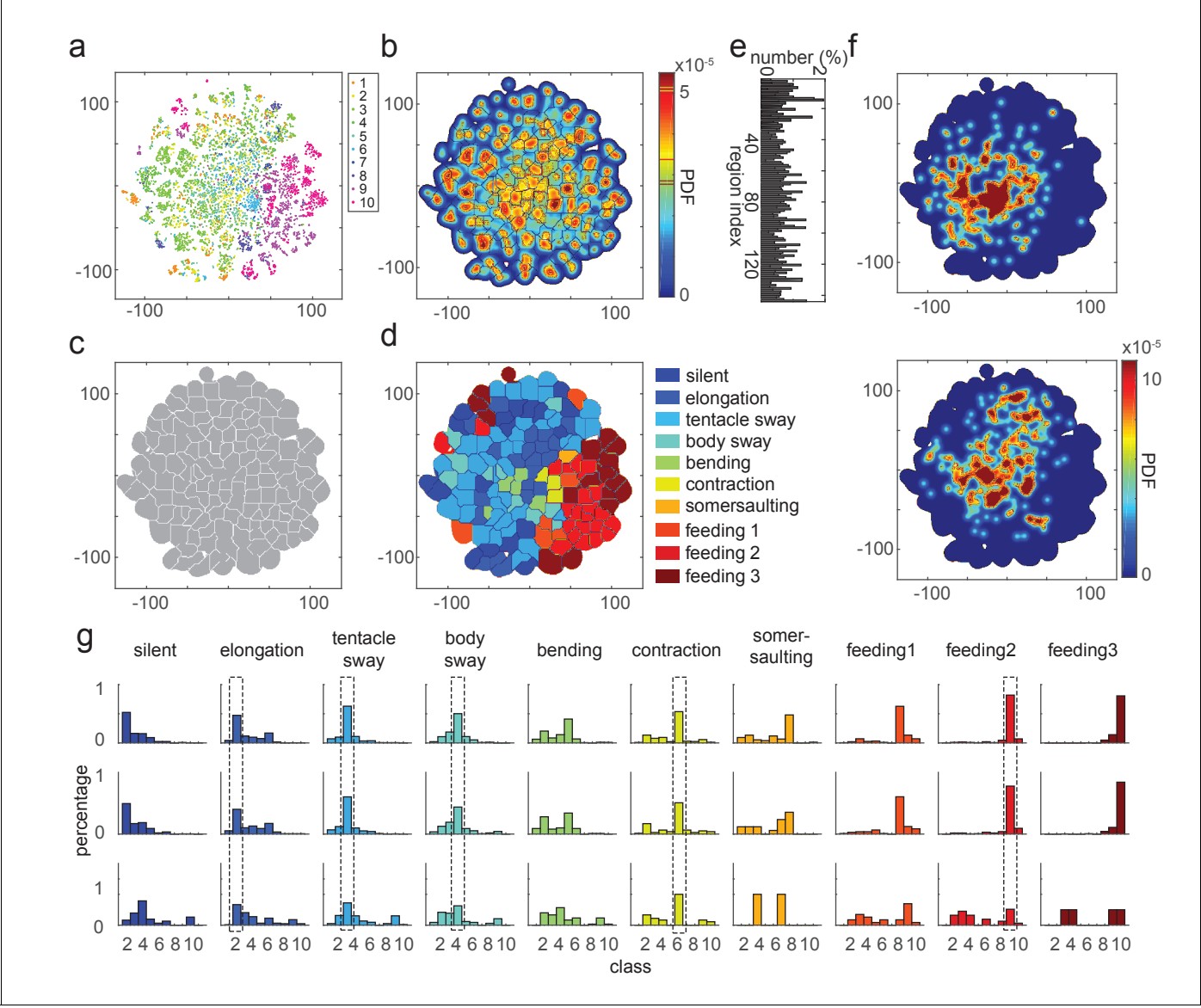

**Figure 4.** t-SNE embedding map of behavior types. (a) Scatter plot with embedded Fisher vectors. Each dot represents projection from a high-dimensional Fisher vector to its equivalent in the embedding space. Color represents the manual label of each dot. (b) Segmented density map generated from the embedding scatter plot. (c) Behavior motif regions defined using the segmented density map. (d) Labeled behavior regions. Color represents the corresponding behavior type of each region. (e) Percentage of the number of samples in each segmented region. (f) Two examples of embedded behavior density maps from test *Hydra* polyps that were not involved in generating the codebooks or generating the embedding space. (g) Quantification of manual label distribution in training, validation and test datasets. Dashed boxes highlight the behavior types that were robustly recognized in all the three datasets. Feeding 1, the tentacle writhing or the first stage of feeding behavior; feeding 2, the ball formation or the second stage of feeding behavior; feeding 3, the mouth opening or the last stage of feeding behavior.

DOI: https://doi.org/10.7554/eLife.32605.020

The following figure supplement is available for figure 4:

**Figure supplement 1.** t-SNE embedding of continuous time windows.

DOI: https://doi.org/10.7554/eLife.32605.021

## Basic behavior of *Hydra* under different experimental conditions

Although basic *Hydra* behaviors such as contraction, feeding and somersaulting have been described for over two centuries, the quantitative understanding of *Hydra* behaviors has been limited by the subjective nature of human annotation and by the amount of data that can be processed

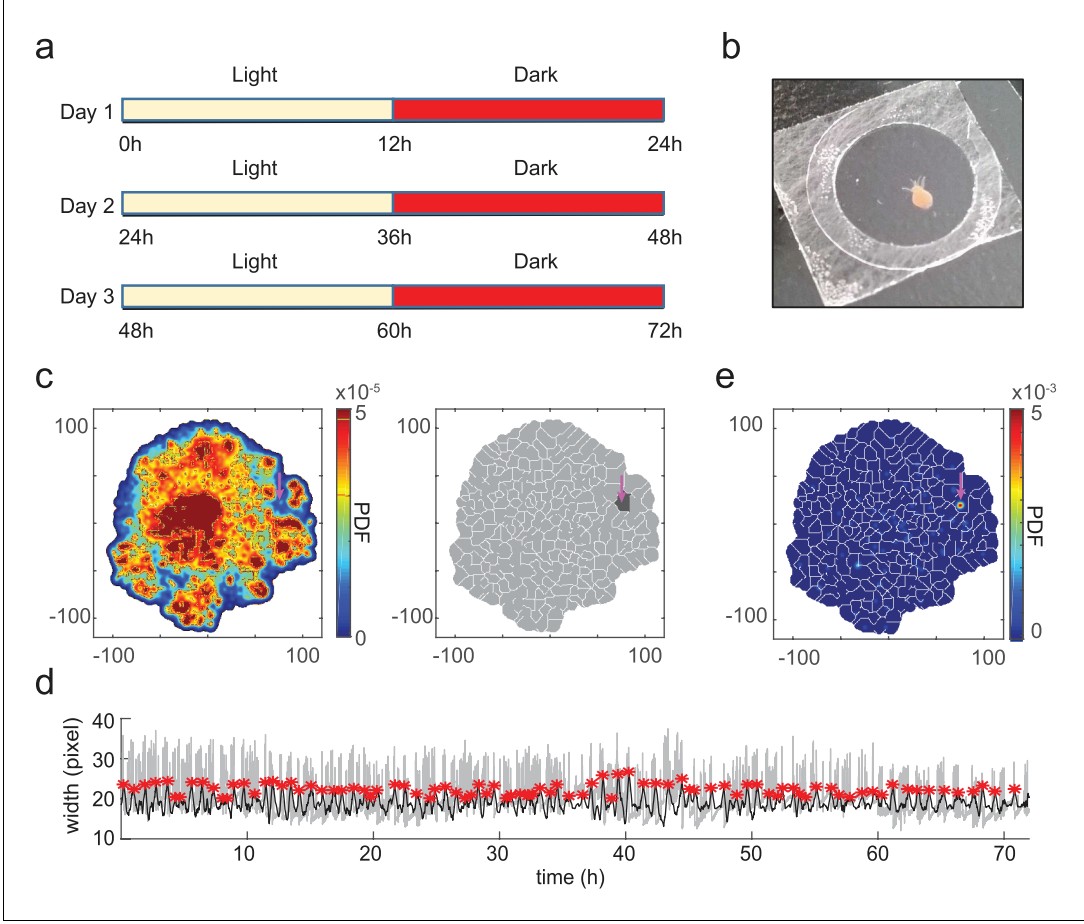

**Figure 5.** t-SNE embedding reveals unannotated egestion behavior. (**a**) Experimental design. A *Hydra* polyp was imaged for 3 days and nights, with a 12 hr light/12 hr dark cycle. (**b**) A *Hydra* polyp was imaged between two glass coverslips separated by a 100 μm spacer. (**c**) Left: density map of embedded behavior during the 3-day imaging. Right: segmented behavior regions with the density map. Magenta arrow indicates the behavior region with discovered egestion behavior. (**d**) Identification of egestion behavior using width profile. Width of the *Hydra* polyp (gray trace) was detected by fitting the body column of the animal to an ellipse, and measuring the minor axis length of the ellipse. The width trace was then filtered by subtracting a 15-minute mean width after each time point from a 15-minute mean width before each time point (black trace). Peaks (red stars) were then detected as estimated time points of egestion events (Materials and methods). (**e**) Density of detected egestion behaviors in the embedding space. Magenta arrow indicates the high density region that correspond to the egestion region discovered in **c**.

DOI: https://doi.org/10.7554/eLife.32605.022

by manual examination. To build quantitative descriptions that link behaviors to neural processes and to explore behavior characteristics of *Hydra*, we used our newly developed method to compare the statistics of behavior under various physiological and environmental conditions.

In its natural habitat, *Hydra* experiences day/night cycles, food fluctuations, temperature variations, and changes in water chemistry. Therefore, we wondered whether *Hydra* exhibit different behavioral frequencies or behavioral variability under dark and light conditions, as well as in starved and well-fed conditions. Since we did not expect *Hydra* to exhibit spontaneous feeding behavior in the absence of prey, we only analyzed six basic behavior types using the trained classifiers: silent, elongation, tentacle swaying, body swaying, bending, and contraction. Lighting conditions (light vs. dark) did not result in any significant changes in either the average time spent in each of the six behavior types (*Figure 6a*) or the individual behavior variability defined by the variation of the percentage of time spent in each behavior in 30 min time windows (*Figure 6b*). Also, compared with starved *Hydra*, well-fed *Hydra* did not show significant changes in the percentage of time spent in

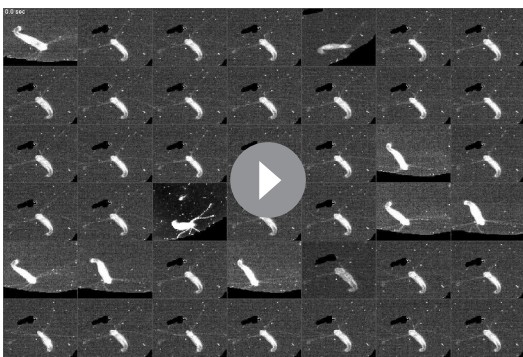

**Video 12.** Examples from the identified silent region in the embedding space.
DOI: https://doi.org/10.7554/eLife.32605.023

elongation behavior (*Figure 6c*), but showed less variability in it (*Figure 6d*; starved: 8.95 ± 0.69%, fed: 5.46 ± 0.53%, p=0.0047).

As *Hydra* polyps vary significantly in size depending on the developmental stage (e.g. freshly detached buds vs. fully grown animals,) and nutrition status (e.g. *Hydra* that has been starved for a week vs. well-fed *Hydra*), we also explored whether *Hydra* of different sizes exhibit different behavioral characteristics. For this, we imaged behaviors of *Hydra* with up to a threefold difference in sizes. Large *Hydra* polyps had similar silent, body swaying, and contraction patterns, but spent slightly less time in elongation, and more in tentacle swaying (*Figure 6e*; elongation small: 22.42 ± 1.35%, large: 17.00 ± 0.74%, p=0.0068; tentacle swaying small: 34.24 ± 1.24%, large: 41.06 ± 2.70%, p=0.03). The individual behavior variability remained unchanged (*Figure 6f*).

Finally, we further inquired if different *Hydra* species have different behavioral repertoires. To answer this, we compared the behaviors of *Hydra vulgaris*, and *Hydra viridissima*, (i.e. green *Hydra*), which contains symbiotic algae in its endodermal epithelial cells(*Martínez et al., 2010*). The last common ancestor of these two species was at the base of *Hydra* radiation. Indeed, we found that *Hydra viridissima* exhibited statistically less silent and bending behaviors, but more elongations (*Figure 6g*; elongation *vulgaris*: 15.74 ± 0.50%, *viridissima*: 18.63 ± 0.87%, p=0.0303; bending *vulgaris*: 2.31 ± 0.27%, *viridissima*: 1.35 ± 0.17%, p=0.0177), while individual *viridissima* specimens also exhibit slightly different variability in bending (*Figure 6h*; *vulgaris*: 2.17% ± 0.26%, *viridissima*: 1.33 ± 0.20%, p=0.0480). We concluded that different *Hydra* species can have different basic behavioral repertoires.

## Discussion

### A machine learning method for quantifying behavior of deformable animals

Interdisciplinary efforts in the emerging field of computational ethology are seeking novel ways to automatically measure and model natural behaviors of animals (*Anderson and Perona, 2014*) (*Berman et al., 2014*; *Branson et al., 2009*; *Brown et al., 2013*; *Creton, 2009*; *Dankert et al., 2009*; *Johnson et al., 2016*; *Kabra et al., 2013*; *Pérez-Escudero et al., 2014*; *Robie et al., 2017*;

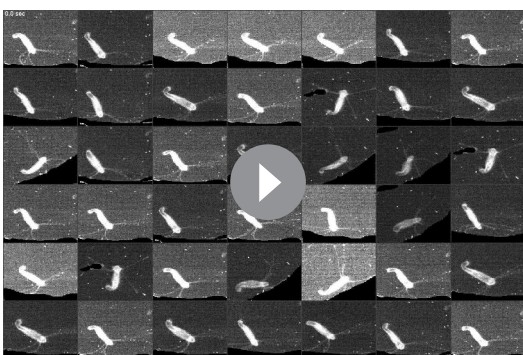

**Video 13.** Examples from the identified slow elongation region in the embedding space.
DOI: https://doi.org/10.7554/eLife.32605.024

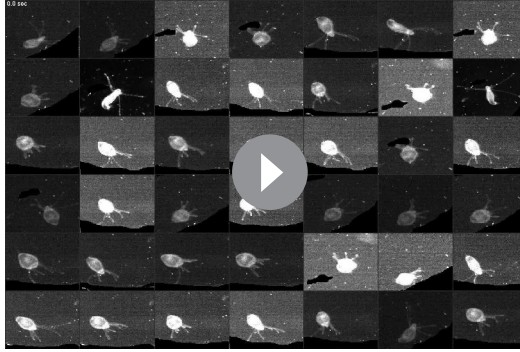

**Video 14.** Examples from the identified fast elongation region in the embedding space.
DOI: https://doi.org/10.7554/eLife.32605.025

Stephens et al., 2008; Swierczek et al., 2011; Wiltschko et al., 2015). Most of these approaches rely on recognizing variation of the shapes of animals based on fitting video data to a standard template of the body of the animal. However, unlike model organisms like worms, flies, fishes and mice, *Hydra* differs dramatically from these bilaterian organisms in having an extremely deformable and elastic body. Indeed, during contraction, *Hydra* appears as a ball with all tentacles shortened, while during elongation, *Hydra* appears as a long and thin column with tentacles relaxed. Moreover, these deformations are non-isometric, that is, different axes, and different parts of the body, change differently. The number of tentacles each *Hydra* has also varies. These present difficult challenges for recognizing *Hydra* behaviors using preset templates.

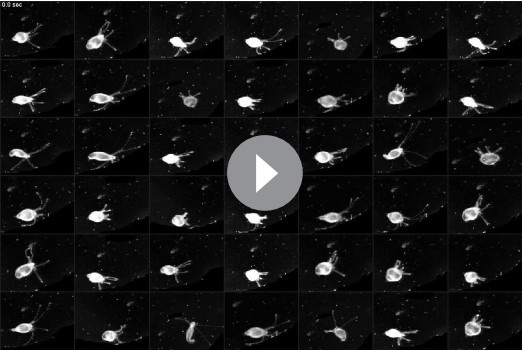

**Video 15.** Examples from the identified inter-contraction elongation region in the embedding space.
DOI: https://doi.org/10.7554/eLife.32605.026

To tackle the problem of measuring behavior in a deformable animal, we developed a novel analysis pipeline using approaches from computer vision that have achieved success in human action classification tasks (*Ke et al., 2007*; *Laptev et al., 2008*; *Poppe, 2010*; *Wang et al., 2009*; *Wang et al., 2011*). Such tasks usually involve various actions and observation angles, as well as occlusion and cluttered background. Therefore, they require more robust approaches to capture stationary and motion statistics, compared to using pre-defined template-based features. In particular, the bag-of-words (BoW) framework is an effective approach for extracting visual information from videos of humans or animals with arbitrary motion and deformation. The BoW framework originated from document classification tasks with machine learning. In this framework, documents are considered 'bags' of words, and are then represented by a histogram of word counts using a common dictionary. These histogram representations are widely used for classifying document types because of their efficiency. In computer vision, the BoW framework considers pictures or videos as 'bags' of visual words, such as small patches in the images, or shape and motion features extracted from such patches. Compared with another popular technique in machine vision, template matching, BoW is more robust against challenges such as occlusion, position, orientation, and viewing angle changes. It also proves to be successful in capturing object features in various scenes, and thus has become one of the most important developments and cutting edge methods in this field. For these reasons, BoW appears ideally suited for the problem behavior recognition tasks of deformable animals, such as *Hydra*.

We modified the BoW framework by integrating other computational methods, including body part segmentation (which introduces spatial information), dense trajectory features (which encode shape and motion statistics in video patches) and Fisher vectors (which represent visual words in a statistical manner). Our choice of framework and parameters proved to be quite adequate, considering both its training and validation accuracy, as well as its generalizability on test datasets (*Figure 2—figure supplement 1*). Indeed, the robust correspondence between supervised, unsupervised and manual classification that we report provides internal cross-validation to the validity and applicability of our BoW machine learning approach. Our developed framework, which uses both supervised and unsupervised techniques, is in principle applicable to all organisms, since it does not rely on specific information of *Hydra*. Compared with previously developed methods, our method would be particularly suitable for

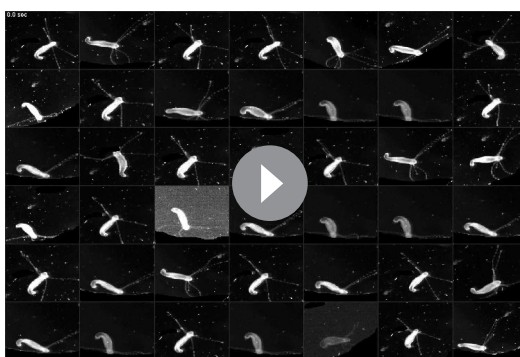

**Video 16.** Examples from the identified bending region in the embedding space.
DOI: https://doi.org/10.7554/eLife.32605.027

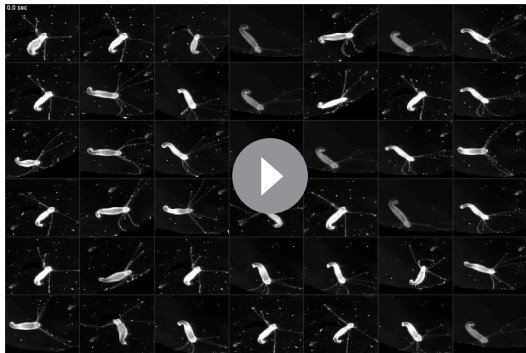

**Video 17.** Examples from the identified tentacle swaying region in the embedding space.
DOI: https://doi.org/10.7554/eLife.32605.028

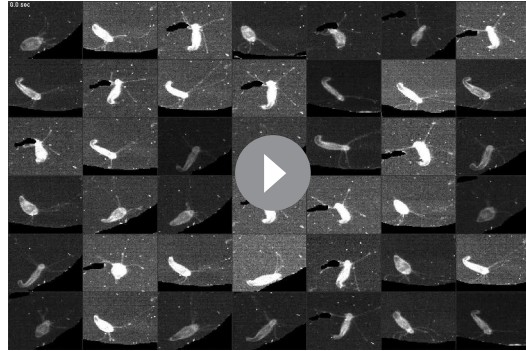

**Video 18.** Examples from the identified initial contraction region in the embedding space.
DOI: https://doi.org/10.7554/eLife.32605.029

behaviors in natural conditions that involve deformable body shapes, as a first step to developing more sophisticated behavioral methods in complex environment for other species.

Our goal was to describe all possible *Hydra* behavior quantitatively. Because of this, we used the BoW framework to capture the overall statistics with a given time frame. We defined the length of basic behavior elements to be 5 s, which maximizes the number of behaviors that were kept intact while uncontaminated by other behavior types (*Figure 1c–d*). However, it should be noted that our approach could not capture fine-level behavior differences, for example, single tentacle behavior. This would require modeling the animal with an explicit template, or with anatomical landmarks, as demonstrated by deformable human body modeling with wearable sensors. Our approach also does not recover transition probabilities between behavior types, or behavioral interactions between individual specimens. In fact, since our method treats each time window as an independent 'bag' of visual words, there was no constraint on the temporal smoothness of classified behaviors. Classifications were allowed to be temporally noisy, therefore they could not be applied for temporal structure analysis. A few studies have integrated state-space models for modeling both animal and human behavior (*Gallagher et al., 2013*; *Ogale et al., 2007*; *Wiltschko et al., 2015*), while others have used discriminative models such as Conditional Random Field models for activity recognition (*Sminchisescu et al., 2006*; *Wang and Suter, 2007*). These methods may provide promising candidates for modeling behavior with temporal structure in combination with our approach (*Poppe, 2010*).

In our analysis pipeline, we applied both supervised and unsupervised approaches to characterize *Hydra* behavior. In supervised classifications (with SVM), we manually defined seven types of behaviors, and trained classifiers to infer the label of unknown samples. In unsupervised analysis (t-SNE), we did not pre-define behavior types, but rather let the algorithm discover the structures that were embedded in the behavior data. In addition, we found that unsupervised learning could discover previously unannotated behavior types such as egestion. However, the types of behaviors discovered by unsupervised analysis are limited by the nature of the encoded feature vectors. Since the BoW model provides only a statistical description of videos, those features do not encode fine differences in behaviors. Due to this difference, we did not apply unsupervised learning to analyze behavior statistics under different environmental and physiological conditions, as supervised learning appeared more suitable for applications where one needs to assign a particular label to a new behavior video.

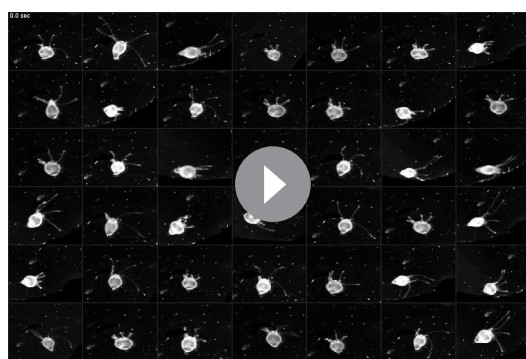

**Video 19.** Examples from the identified contracted contraction region in the embedding space.
DOI: https://doi.org/10.7554/eLife.32605.030

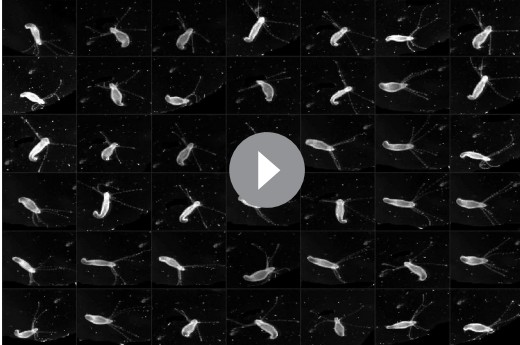

**Video 20.** Examples from the identified egestion region in the embedding space.
DOI: https://doi.org/10.7554/eLife.32605.031

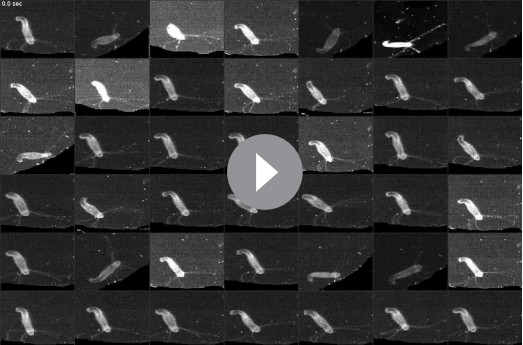

**Video 21.** Examples from the identified hypostome movement region in the embedding space.
DOI: https://doi.org/10.7554/eLife.32605.032

## Stability of the basic behavioral repertoire of *Hydra*

Once we established the reliability or our method, we quantified the differences between six basic behaviors in *Hydra* under different experimental conditions with two different species of *Hydra* and found that *Hydra vulgaris* exhibits essentially the same behavior statistics under dark/light, large/ small and starved/fed conditions. Although some small differences were observed among experimental variables, the overall dwell time and variance of the behavioral repertoire of *Hydra* were unexpectedly very similar in all these different conditions. Although we could not exclude the possibility that there were differences in the transition probabilities between behaviors, our results still

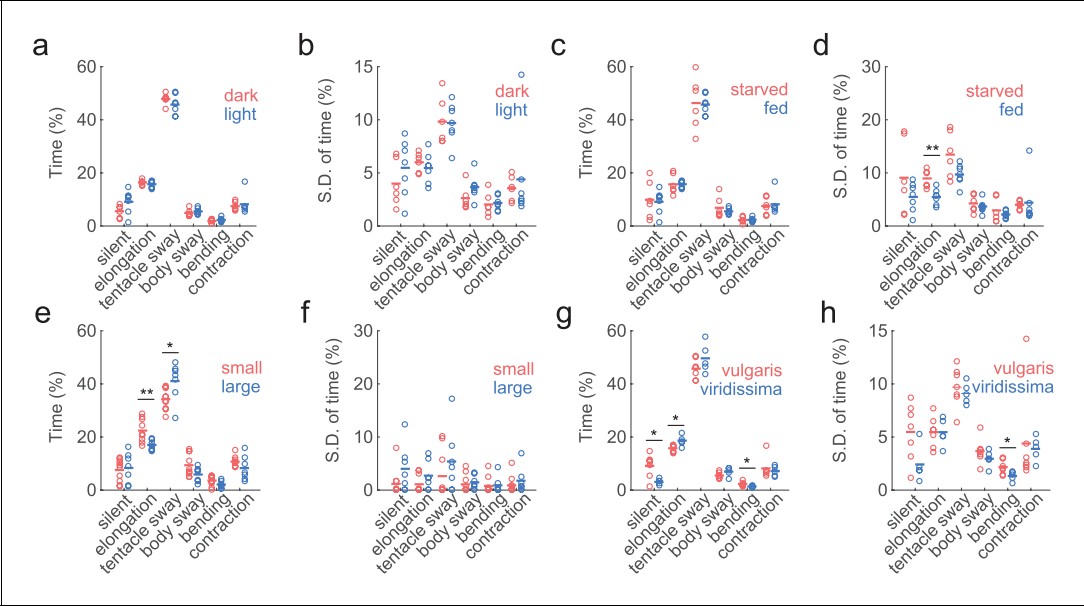

**Figure 6.** Similar behavior statistics under different conditions but differences across species. (**a**) Percentage of time *Hydra* spent in each behavior, in dark (red to infra-red) and light conditions. Each circle represents data from one individual. The horizontal line represents the average of all samples. Red represents dark condition, blue represents light condition. ($n_{dark}$ = 6, $n_{light}$ = 7) (**b**) Standard deviations of behaviors within each individual animal, calculated with separate 30 min time windows in the recording. Each circle represents the behavior variability of one individual. (**c**) Percentage of time *Hydra* spent in each behavior, in starved and well-fed condition. ($n_{starved}$ = 6, $n_{fed}$ = 7) (**d**) Standard deviations of individual behaviors under starved and well-fed conditions. (**e**) Percentage of time small and large *Hydra* spent in each behavior. ($n_{small}$ = 10, $n_{large}$ = 7). (**f**) Standard deviations of behaviors of small and large individuals. (**g**) Percentage of time *Hydra vulgaris* and *Hydra viridissima* spent in each behavior type. ($n_{vulgaris}$ = 7, $n_{viridissima}$ = 5). (**h**) Standard deviations of individual brown and green *Hydra*. *$p < 0.05$, **$p < 0.01$, Wilcoxon rank-sum test.
DOI: https://doi.org/10.7554/eLife.32605.033

show that , from the six basic behaviors analyzed, *Hydra* possess a surprisingly robust behavioral frequencies and similarities across environmental and physiological conditions, while interspecies differences introduce stronger behavior differences.

*Passano and McCullough (1964)* reported that *Hydra littoralis*, a close relative with our *Hydra vulgaris* AEP strain (*Martínez et al., 2010*), showed fewer contraction bursts in the evenings and nights than in the day, and feeding every third or fourth day resulted in fewer contraction bursts than was seen with daily feeding. However, they detected contraction bursts by electrical recording of epithelial cell activity, and defined coordinated activity as a contraction event. In our method, we did not measure the number of such events, but instead measured the number of time windows that contain such contractile behavior. This is essentially a measurement of the time spent in contractions instead of frequency of individual events. Using natural light instead of lamp light could also lead to a difference in the observation results. Interestingly, we observed that *Hydra vulgaris* exhibits different behavior statistics compared with *Hydra viridissima*. The split leading to *Hydra vulgaris* and *Hydra viridissima* is the earliest one in the *Hydra* phylogenetic tree (*Martínez et al., 2010*), thus these two species are quite divergent. *Hydra viridissima* also possesses symbiotic algae, and requires light for normal growth (*Lenhoff and Brown, 1970*). These differences in genetics and growth conditions could help explaining the observed behavioral differences.

Given the similarity in statistics of basic behaviors in different conditions across different animals within the same species, we naturally wondered if our approach might not be effective or sensitive enough to detect significant behavioral differences among animals. However, the high accuracy of the classification of annotated behavior subtypes (*Figure 3*) and also the method reproducibility, with small variances when measuring different datasets, rules out the possibility that this machine learning method is insensitive, in which case the results of our behavioral analysis would have been noisy and irreproducible. This conclusion was corroborated by the statistical differences in behavior found across two different *Hydra* species.

We had originally expected to observe larger variability of behaviors under different experimental conditions and we report essentially the opposite result. We interpret the lack of behavioral differences across individuals as evidence for robust neural control of a basic behavioral pattern, which appears unperturbed by different experimental conditions. While this rigidity may not seem ideal if one assumes that behavior should flexibly adapt to the environment, it is possible that the six behaviors we studied represent a basic 'house keeping' repertoire that needs to be conserved for the normal physiology and survival of the animal. Our results are reminiscent of work on the stomatogastric ganglion of crustaceans that has revealed homeostatic mechanisms that enable central pattern generators to function robustly in different environmental conditions, such as changes in temperature (*Haddad and Marder, 2017*). In fact, in this system, neuropeptides and neuromodulators appear to be flexibly used to enable circuit and behavioral homeostasis (*Marder, 2012*). Although we do not yet understand the neural mechanisms responsible for the behavioral stability in *Hydra*, it is interesting to note that the *Hydra* genome has more than one hundred neuropeptides that could play neuromodulator roles (*Chapman et al., 2010*; *Fujisawa and Hayakawa, 2012*). This vast chemical toolbox could be used to supplement a relatively sparse wiring pattern with mechanisms to ensure that the basic behavior necessary for the survival of the animal remains constant under many different environmental conditions. One can imagine that different neuromodulators could alter the biophysical properties of connections in the *Hydra* nerve net and thus keep a stable operating regime of its neurons in the physiological states.

In addition, a possible reason for the behavioral similarity among different specimens of *Hydra* could be their genetic similarities. We used animals derived from the same colony (*Hydra* AEP strain), which was propagated by clonal budding. Thus, it is likely that many of the animals were isogenic, or genetically very similar. The lack of genetic variability, although it does not explain the behavioral robustness, could partly be a reason behind our differences across species, and it would explain a relatively small quantitative variability across animals of our *H. vulgaris* colony, as opposed to a larger variability in specimens from the wild.

Finally, it is also possible that the behavioral repertoire of cnidarians, which represents some of the simplest nervous systems in evolution in structure and probably also in function, could be particularly simple and hardwired as compared with other metazoans or with bilaterians. From this point of view, the robustness we observed could reflect a 'passive stability' where the neural mechanisms are simply unresponsive to the environment, as opposed to a homeostatic 'active stability',

generated perhaps by neuromodulators. This distinction mirrors the difference between open-loop and closed-loop control systems in engineering (*Schiff, 2012*). Thus, it would be fascinating to reverse engineer the *Hydra* nerve net and discern to what extent its control mechanisms are regulated externally. Regardless of the reason for this behavioral stability, our analysis provides a strong baseline for future behavioral analysis of *Hydra* and for the quantitative analysis of the relation between behavior, neural and non-neuronal cell activity.

## Hydra as a model system for investigating neural circuits underlying behavior

Revisiting *Hydra* as a model system with modern imaging and computational tools to systematically analyze its behavior provides a unique opportunity to image the entire neural network in an organism and decode the relation between neural activity and behaviors (*Bosch et al., 2017*). With recently established GCaMP6s transgenic *Hydra* lines (*Dupre and Yuste, 2017*) and the automated behavior recognition method introduced in this study, it should now be possible to identify the neural networks responsible for each behavior in *Hydra* under laboratory conditions.

With this method, we demonstrate that we are able to recognize and quantify *Hydra* behaviors automatically, and to identify novel behavior types. This allows us to investigate the behavioral repertoire stability under different environmental, physiological and genetic conditions, providing insight into how a primitive nervous system adapt to its environment. Although our framework does not currently model temporal information directly, it serves as a stepping-stone toward building more comprehensive models of *Hydra* behaviors. Future work that incorporates temporal models would allow us to quantify behavior sequences, and to potentially investigate more complicated behaviors in *Hydra* such as social and learning behaviors.

As a member of the phylum Cnidaria, *Hydra* is a sister to bilaterians, and its nervous system and bilaterians nervous systems share a common ancestry. As demonstrated by the analysis of its genome (*Chapman et al., 2010*), *Hydra* is closer in gene content to the last common ancestor of the bilaterian lineage than some other models systems used in neuroscience research, such as *Drosophila* and *C. elegans*. In addition, comparative studies are essential to discern whether the phenomena and mechanisms found when studying one particular species are specialized or general and can thus help illuminate essential principles that apply widely. Moreover, as was found in developmental biology, where the body plan of animals is built using the same logic and molecular toolbox (*Nüsslein-Volhard and Wieschaus, 1980*), it is possible that the function and structure of neural circuits could also be evolutionarily conserved among animals. Therefore, early diverging metazoans could provide an exciting opportunity to understand the fundamental mechanisms by which nervous systems generate and regulate behaviors.

## Materials and methods

### Hydra behavior dataset

The *Hydra* behavior dataset consisted of 53 videos from 53 *Hydra* with an average length of 30 min. The AEP strain of *Hydra* was used for all experiments. *Hydra* polyps were maintained at 18°C in darkness and were fed with *Artemia* nauplii once or more times a week by standard methods (*Lenhoff and Brown, 1970*). During imaging, *Hydra* polyps were placed in a 3.5 cm plastic petri dish under a dissecting microscope (Leica M165) equipped with a sCMOS camera (Hamamatsu ORCA-Flash 4.0). Videos were recorded at 5 Hz. *Hydra* polyps were allowed to behave either undisturbed, or in the presence with reduced L-glutathione (Sigma-Aldrich, G4251-5G) to induce feeding behavior, since *Hydra* does not exhibit feeding behavior in the absence of prey.

### Manual annotation

Each video in the *Hydra* behavior dataset was examined manually at a high playback speed, and each frame in the video was assigned a label in the following eleven classes based on the behavior that *Hydra* was performing: silent, elongation, tentacle swaying, body swaying, bending, contraction, somersaulting, tentacle writhing of feeding, ball formation of feeding, mouth opening of feeding, and a none class. These behaviors were labeled as 1 through 11, where larger numbers correspond to more prominent behaviors, and the none class is labeled as 0. To generate manual labels for a

given time window, the top two most frequent labels, $L_1$ and $L_2$, within this time window were identified. The window was assigned as $L_2$ if its count exceed $L_1$ by three-fold and if $L_1$ is more prominent than $L_2$; otherwise, the window was assigned as $L_1$. This annotation method labels time windows as more prominent behaviors if behaviors with large motion, e.g. contraction, happens in only a few frames, while the majority of frames are slow behaviors.

## Video pre-processing

Prior work has shown that the bag of words methods for video action classification perform better when encoding spatial structure (*Taralova et al., 2011*; *Wang et al., 2009*). Encoding spatial information is especially important in our case because allowing the animal to move freely produces large variations in orientation, which is not related to behavior classification. Therefore, we performed a basic image registration procedure that keeps the motion information invariant, but aligns the *Hydra* region to a canonical scale and orientation. This involves three steps: background segmentation, registration, and body part segmentation. In brief, the image background was calculated by a morphological opening operation, and the background was removed from the raw image. Then, image contrast was adjusted to enhance tentacle identification. Images were then segmented by clustering the pixel intensity profiles to three clusters corresponding to *Hydra* body, weak-intensity tentacle regions and background by k-means, and the largest cluster from the result was treated as background, and the other two clusters as foreground, that is *Hydra* region. Connected components that occupied less than 0.25% of total image area in this binary image were removed as noise, and the resulting *Hydra* mask was then dilated by three pixels. To detect the body column, the background-removed image was convolved with a small 3-by-3 Gaussian filter with sigma equals one pixel, and the filtered image was thresholded with Otsu's segmentation algorithm. The binarization was repeated with a new threshold defined with Otsu's method within the previous above-threshold region, and the resulting binary mask was considered as the body column. The body column region was then fitted with an ellipse; the major axis, centroid, and orientation of the ellipse were noted. To determine the orientation, two small square masks were placed on both ends of the ellipse along the major axis, and the area of the *Hydra* region excluding the body column under the patch was calculated; the end with the larger area was defined as the tentacle/mouth region, and the end with the smaller area was defined as the foot region. To separate the *Hydra* region into three body parts, the part under the upper body square mask excluding the body column was defined as the tentacle region, and the rest of the mask was split at the minor axis of the ellipse; the part close to the tentacle region was defined as the upper body region, and the other as the lower body region. This step has shown to improve representation efficiency (*Figure 2—figure supplement 1b*).

Each 5-s video clip was then centered by calculating the average ellipse centroid position and centering it. The average major axis length and the average orientation were also calculated. Each image in the video clip was rotated according to the average orientation to make the *Hydra* vertical, and was scaled to make the length of the *Hydra* body 100 pixels, with an output size of 300 by 300 pixels, while only keeping the region under the *Hydra* binary mask.

## Feature extraction

Video features including HOF, HOG and MBH were extracted using a codebase that was previously released (*Wang et al., 2011*). Briefly, interest points were densely sampled with five pixels spacing at each time point in each 5 s video clip and were then tracked throughout the video clip with optical flow for 15 frames. The tracking quality threshold was set to 0.01; the minimum variation of trajectory displacement was set to 0.1, the maximum variation was set to 50, and the maximum displacement was set to 50. The neighboring 32 pixels of each interest point were then extracted, and HOF (8 dimensions for eight orientations plus one extra zero bin), HOG (eight dimensions) and MBH (eight dimensions) features were calculated with standard procedures. Note that MBH was calculated for horizontal and vertical optical flow separately, therefore two sets of MBH features, MBHx and MBHy were generated. All features were placed into three groups based on the part of body they fall in, that is tentacles, upper body column, and lower body column. All parameters above were cross-validated with the training and test datasets.

## Gaussian mixture codebook and Fisher vector

A Gaussian mixture codebook and Fisher vectors were generated using the code developed by Jegou et al. for each feature type (*Jégou et al., 2012*), using 50 *Hydra* in the behavior dataset that includes all behavior types. Features from each body part were centered at zero, then PCA was performed on centered features from all three body parts, keeping half of the original dimension (five for HOF, four for HOG, MBHx and MBHy). Whitening was performed on the PCA data as following, which de-correlates the data and removes redundant information:

$$x_{\text{white},\,i} = \frac{x_i}{\sqrt{\lambda_i}}$$

where $x$ denotes principal components, and $\lambda$ denotes eigenvalues. $K = 256$ Gaussian mixtures were then fitted with the whitened data using a subset of 256,000 data points. We then calculated the Fisher vectors as following:

$$z_X = L_\lambda \, \nabla_\lambda \, \mathrm{L}(\mathrm{X}|\lambda)$$

where $X = \{x_t,\ t = 1 \dots T\}$ is a set of $T$ data points that were assumed to be generated with Gaussian distributions $u_\lambda(x) = \sum_{i=1}^{K} w_i u_i(x)$, with $\lambda = \{w_i, \mu_i, \sigma_i,\ i = 1, \dots, K\}$ denotes the Gaussian parameters, and $L_\lambda$ is the decomposed Fisher Information Matrix:

$$F_\lambda^{-1} \equiv E_{x \sim u_\lambda}\left[ \nabla_\lambda \log u_\lambda(x) \nabla_\lambda \log u_\lambda(x)^{\mathrm{T}} \right] = L_\lambda^{\mathrm{T}} L_\lambda$$

Fisher vectors then represent the normalized gradient vector obtained from Fisher kernel $K(X, X')$:

$$K(X, X') = \nabla_\lambda \, \mathrm{L}(\,\mathrm{X}\,|\,\lambda\,)^{\mathrm{T}} \, F_\lambda^{-1} \, \nabla_\lambda \, \mathrm{L}(\,\mathrm{X}'\,|\,\lambda\,) = z_X^T z_X$$

Comparing with hard-assigning each feature to a code word, the Gaussian mixtures can be regarded as probabilistic vocabulary, and Fisher vectors encode information of both the position and the shape of each word with respect to the Gaussian mixtures. Power normalization was then performed on the Fisher vectors to improve the quality of representation:

$$f(z) = \text{sign}(z)|z|^\alpha$$

with $\alpha = 0.5$, followed by $l_2$ normalization, which removes scale dependence (*Perronnin et al., 2010*). The final representation of each video clip is a concatenation of Fisher vectors of HOF, HOG, MBHx and MBHy. In this paper, the GMM size was set to 128 with cross-validation (*Figure 2—figure supplement 1c*).

## SVM classification

PCA was first performed on the concatenated Fisher vectors to reduce the dimensions while keeping 90% of the original variance. A random 90% of samples from the 50 training *Hydra* were selected as training data, and the remaining 10% were withheld as validation data. Another three *Hydra* that exhibit all behavior types were kept as test data. Because each behavior type has different numbers of data points, we trained SVM classifiers using the libSVM implementation (*Chang and Lin, 2011*) by assigning each type a weight of $w_i = \left(\sum_i N_i\right)/N_i$, where $i = 1, \dots, 7$ denotes the behavior type, and $N_i$ denotes the number of data points that belong to type $i$. We trained SVM classifiers with a radial basis kernel, allowing probability estimate, and a fivefold cross-validation testing the cost parameter $c$ with a range of $\log_2 c \in \{-5{:}2{:}15\}$, and the $g$ in the kernel function with a range of $\log_2 g \in \{-5{:}2{:}15\}$, where $\{-5{:}2{:}15\}$ denotes integers ranging from $-5$ to 15 with a step of 2. The best parameter combination from cross-validation was chosen to train the SVM classifiers.

To classify test data, features were extracted as above and were encoded with Fisher vectors with the codebook generated from the training data. PCA was performed using the projection matrix from training data. A probability estimate for each behavior type was given by the classifiers, and the final assigned label is the classifier with the highest probability. For soft classifications, we allowed up to three labels for each sample if the second highest label probability is >50% of the highest label, and the third is >50% of the second highest label. To evaluate classification

performance, true positives (TP), false positives (FP), true negatives (TN) and false negatives (FN) were calculated. Accuracy was defined as $\mathrm{Acc} = (\mathrm{TP} + \mathrm{TN})/(\mathrm{TP} + \mathrm{TN} + \mathrm{FP} + \mathrm{FN})$; precision was defined as $\mathrm{Prc} = \mathrm{TP}/(\mathrm{TP} + \mathrm{FP})$; recall was defined as $\mathrm{Acc} = \mathrm{TN}/(\mathrm{TN} + \mathrm{FP})$. Two other measurements were calculated: true positive rate $\mathrm{TPR} = \mathrm{TP}/(\mathrm{TP} + \mathrm{FN})$, and false-positive rate $\mathrm{FPR} = \mathrm{FP}/(\mathrm{FP} + \mathrm{TN})$. Plotting TPR against FPR gives the standard ROC curve, and the area under curve (AUC) reflects the performance of classification. In this plot, a straight line TPR = FPR with AUC = 0.5 represents random guess; the upper left quadrant with AUC >0.5 represents better performance than random.

### t-SNE embedding

Embedding was performed with the dimension-reduced data. A random 80% of the dataset from the 50 training *Hydra* were chosen to generate the embedding map, and the remaining 20% were withheld as validation dataset. Three other *Hydra* were used as test dataset. We followed the procedures of *Berman et al. (2014)*, with a slight modification that uses Euclidean distance as the distance measurement. Embedding perplexity was chosen as 16. To generate a density map, a probability density function was calculated in the embedding space by convolving the embedded points with a Gaussian kernel; $\sigma$ of the Gaussian was chosen to be 1/40 of the maximum value in the embedding space by cross-validation with human examination to minimize over-segmentation. In the 3-day dataset, $\sigma$ was chosen to be 1/60 of the maximum value in order to reveal finer structures. To segment the density map, peaks were found in the density map, a binary map containing peak positions was generated, and peak points were dilated by three pixels. A distance map of the binary image was generated and inverted, and the peak positions were set to be minimum. Watershed was performed on the inverted distance map, and the boundaries were defined with the resulting watershed segmentation.

### Egestion detection

Estimated egestion time points were calculated by first extracting the width profile of *Hydra* from the pre-processing step, then filtering the width profile by taking the mean width during 15 min after each time point $t$, and the mean width during 15 min before time $t$, and subtracting the former from the latter. Peaks were detected on the resulting trace and were regarded as egestion behaviors, since they represent a sharp decrease in the thickness of the animals.

### Behavior experiments

All *Hydra* used for experiments were fed three times a week and were cultured at 18°C. On non-feeding days, the culture medium was changed. *Hydra viridissima* was cultured at room temperature under sunlight coming through the laboratory windows. For imaging, animals were placed in a petri dish under the microscope without disturbance to habituate for at least 30 min. Imaging typically started between 7 pm and 9 pm, and ended between 9 am and 11 am except for the large/small experiments. All imagings were done excluding environmental light by putting a black curtain around the microscope. For dark condition, a longpass filter with a cutoff frequency of 650 nm (Thorlabs, FEL0650) was placed at the source light path to create '*Hydra* darkness' (*Passano and McCullough, 1962*). For starved condition, *Hydra* were fed once a week. For the large/small experiment, *Hydra* buds that were detached from their parents within 3 days were chosen as small *Hydra*, and mature post-budding mature *Hydra* polyps were chosen as large *Hydra*. There was a two- to three-fold size difference between small and large *Hydra* when they were relaxed. However, since the *Hydra* body was constantly contracting and elongating, it was difficult to measure the exact size. Imaging for this experiment was done during the day time for 1 hr per *Hydra*.

### Statistical analysis

All statistical analyses were done using Wilcoxon rank-sum test unless otherwise indicated. Data is represented by mean ± S.E.M unless otherwise indicated.

### Resource availability

The code for the method developed in this paper is available at https://github.com/hanshuting/Hydra_behavior. A copy is archived at https://github.com/elifesciences-publications/hydra_behavior

(*Han, 2018b*). The annotated behavior dataset is available on Academic Commons (dx.doi.org/10.7916/D8WH41ZR).

## Acknowledgements

We thank Drs. Robert Steele, Charles David, and Adrienne Fairhall for discussions. This material is based upon work supported by the Defense Advanced Research Projects Agency (DARPA) under Contract No. HR0011-17-C-0026. SH is a Howard Hughes Medical Institute International Student Research fellow. This work was partly supported by the Grass Fellowship (CD) during the summer of 2016, and CD would like to thank the Director, Associate Director, members of the Grass laboratory and Grass Foundation for their generous feedback and support. RY was a Whitman fellow at the Marine Biological Laboratory and this Hydra research was also supported in part by competitive fellowship funds from the H Keffer Hartline, Edward F MacNichol, Jr. Fellowship Fund, and the E E Just Endowed Research Fellowship Fund, Lucy B. Lemann Fellowship Fund, and Frank R. Lillie Fellowship Fund of the Marine Biological Laboratory in Woods Hole, MA. The authors declare no competing financial interests.

## Additional information

### Funding

| Funder | Grant reference number | Author |
| --- | --- | --- |
| Defense Advanced Research Projects Agency | HR0011-17-C-0026 | Rafael Yuste |
| Howard Hughes Medical Institute | Howard Hughes Medical Institute International Student Research Fellowship | Shuting Han |
| Grass Foundation | Grass Fellowship | Christophe Dupre |

The funders had no role in study design, data collection and interpretation, or the decision to submit the work for publication.

### Author contributions

Shuting Han, Conceptualization, Resources, Data curation, Software, Formal analysis, Investigation, Visualization, Methodology, Writing—original draft, Writing—review and editing; Ekaterina Taralova, Resources, Methodology, Writing—review and editing; Christophe Dupre, Resources, Data curation, Investigation, Writing—review and editing; Rafael Yuste, Conceptualization, Funding acquisition, Writing—review and editing

### Author ORCIDs

Shuting Han http://orcid.org/0000-0001-9315-3089
Christophe Dupre https://orcid.org/0000-0002-5929-8492
Rafael Yuste https://orcid.org/0000-0003-4206-497X

### Decision letter and Author response

Decision letter https://doi.org/10.7554/eLife.32605.038
Author response https://doi.org/10.7554/eLife.32605.039

## Additional files

### Major datasets

The following dataset was generated:

| Author(s) | Year | Dataset title | Dataset URL | Database, license, and accessibility information |
|-----------|------|---------------|-------------|--------------------------------------------------|
| Han S | 2018 | Hydra behavior dataset | http://dx.doi.org/10.5061/dryad.f6v067r | Available at Dryad Digital Repository under a CC0 Public Domain Dedication. |

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
