## [Decision Letter]

Thank you for submitting your article "Comprehensive machine learning analysis of *Hydra* behavior reveals a stable behavioral repertoire" for consideration by *eLife*. Your article has been reviewed by three peer reviewers,, one of whom, Ronald Calabrese is a member of our Board of Reviewing Editors and the evaluation has been overseen by Eve Marder as the Senior Editor. The following individual involved in review of your submission has also agreed to reveal his identity: Gordon J Berman.

The reviewers have discussed the reviews with one another and the Reviewing Editor has drafted this decision to help you prepare a revised submission

Summary:

This is an interesting manuscript that reports the development a novel behavioral analysis pipeline using approaches from computer vision to assess various natural behaviors compatible with variable observation angles. It extracts visual information from videos of animals with arbitrary motion and deformation. It then integrates a few modern computational methods, including dense trajectory features, Fisher vectors and t-SNE embedding for robust recognition and classification of Hydra behaviors with the bag-of-words framework. The pipeline, which uses both supervised and unsupervised techniques, is suitable for use not only with Hydra, as demonstrated here, but compared with previously developed methods this method is particularly suitable for behaviors in natural conditions that involve animals with deformable body shapes. The pipeline is used to describe behaviors in Hydra and is successful in identifying previously-identified behaviors and novel behaviors not previously identified. The paper then goes on to specify the frequency and variance of these behaviors under a variety of conditions (e.g. fed vs. unfed) and surprisingly found similar behavioral statistics. They conclude that the behavioral repertoire of Hydra is robust which may reflect homeostatic neural principles or a particularly stable ground state of the nervous system. Comparisons with another distantly related Hydra species interestingly reveal some strong differences.

Essential revisions:

The reviewers found that there was a substantial contribution to methodology and behavioral analyses of Hydra in this paper but had concerns that these contributions were obscured by the explanation of the methodology and the presentation and interpretation of the behavioral results. There concerns are well summarized in the thorough review discussion. Reviewer #2 commented "I agree with the importance of quantifying Hydra behavior but have two reservations:

1) The choices for their particular pipeline need to be clarified and discussed. This includes Reviewer #3's concern on the 5-second window but extends to other choices in the stream. As machine learning techniques advance it is getting much easier to represent video information with numbers and I expect that as a result we will see many future advances in behavioral representation. However, these representations are often idiosyncratic and so it is important to understand what aspects are universal, or at the very least include a discussion about various choices. I also think it is important to discuss what kind of behavior might be missing in this approach.

2) They need to do a better job of motivating and discussing the important questions that their quantitative behavioral repertoire can answer. This is the science of behavior not simply representation of videos as numbers. And it's here that we can learn how Hydra compares to worms and flies what we might expect to find in the neural recordings."

Reviewer #3 then commented "I think that Reviewer #2 put it well. I would like to see them talk a bit more about:

1) The motivations for their representational choices and, importantly, the consequences and implications that these choices have.

2) How they anticipate using these numbers to answer biological questions. Any measurement representation should have a rational relationship to the questions being asked, so addressing what their method will be useful for (and what it won't be useful for) will be valuable for the literature moving forward. Will this simply be a better detection technique, or does their unsupervised approach allow the field fundamentally new types of behavioral measurements?"

These concerns and the minor issues from the original review that are appended can form the basis of a revision that clarifies the methodology and brings out the behavioral significance revealed by the new methodology.

[Editors' note: further revisions were requested prior to acceptance, as described below.]

Thank you for resubmitting your work entitled "Comprehensive machine learning analysis of Hydra behavior reveals a stable behavioral repertoire" for further consideration at *eLife*. Your revised article has been favorably evaluated by Eve Marder (Senior Editor), a Reviewing editor, and three reviewers.

The manuscript has been improved but there are some remaining issues that need to be addressed before acceptance, as outlined below:

The authors have endeavored to address many of the major concerns of the previous review with further explanations of the method and its underlying assumptions and method choices, and the inherent limitations of the method/analysis. They have also tried to clarify how their method can be used to answer biological questions. There is still one major concern.

1a) The authors should try windows greater than 5 seconds. It's hardly surprising that less than five seconds is less effective, but why not 8, 10, 20? Just saying "we noticed 5 seconds is a reasonable length to define a single behavior" is hardly convincing (neither is Figure 1C).

1b) It still remains possible that the highly-fragmented t-SNE representation results form the fact that behaviors are unnecessarily chopped-up by imposing a 5 second window. Problems might occur because the behavior spans a window boundary. The analysis should be performed using a sliding 5-second window rather than separated windows. This may remove some of the observed over-segmentation of the space. There are several methods (including one in Berman, 2014, but others as well) for handling tens to hundreds of millions of data points. Since the space is one of the cruxes of the paper's arguments, and the authors might get better results with the sliding window, it seems somewhat remiss to not attempt this (it would be ~ 24 hours of running time on a machine that can handle a 30,000-point t-SNE). The Barnes-Hut implementation from: https://lvdmaaten.github.io/tsne/ may prove helpful.

---

## [Author Response]

1) The choices for their particular pipeline need to be clarified and discussed. This includes Reviewer #3's concern on the 5-second window but extends to other choices in the stream. As machine learning techniques advance it is getting much easier to represent video information with numbers and I expect that as a result we will see many future advances in behavioral representation. However, these representations are often idiosyncratic and so it is important to understand what aspects are universal, or at the very least include a discussion about various choices. I also think it is important to discuss what kind of behavior might be missing in this approach.

Our goal was to describe all possible *Hydra* behavior quantitatively. For this purpose, we chose the bag-of-words (BoW) framework, which captures the overall statistics of a dataset with a given time frame and has demonstrated success in deformable human action classification tasks. The BoW framework originated from document classification tasks in the machine learning field. In this framework, documents are considered “bags” of words, and are then represented by a histogram of word counts using a common dictionary. These histogram representations are demonstrated to be efficient for classifying document types. In computer vision, the BoW framework considers instead pictures or videos as “bags” of visual words such as small patches in the images, or shape and motion features extracted from such patches. Compared with another popular technique, template matching, BoW is more robust against challenges such as occlusion, position, orientation, and viewing angle changes. It also proves to be successful in capturing object features in various scenes and is thus one of the most important concepts and cutting-edge techniques in this field. For behavior recognition tasks of deformable animals, BoW is therefore ideally suited for the problem.

In the BoW framework, we made the choices of segmenting the *Hydra* from background, scale and register *Hydra* to eliminate variance introduced by size and orientation, using dense trajectories features that captures both shape and motion statistics, using Gaussian Mixture codebooks and Fisher vectors to encode the features in a probabilistic way, classifying *Hydra* behaviors with standard SVM classifiers, and identifying behavior types with the t-SNE embedding which has demonstrated success in fly behavior analysis in an unsupervised way. Although these choices may seem arbitrary, they are anchored on the structure of the data and task at hand, as we explain below. Our developed framework is a modified version of the original BoW framework, which is simply a normalized histogram representation of selected visual features. This modification includes the key steps of 3 body part segmentation, dense trajectory features, and Fisher vector encoding. We also compared the supervised classification performance of histogram representation vs. Fisher vector representation, the effect of introducing body part segmentation of 3 and 6 segments (Figure 2—figure supplement 1B), different time window sizes (Figure 2—figure supplement 1A), and different Gaussian Mixture codebook sizes (Figure 2—figure supplement 1C). Our choice of framework and parameters proves to be quite ideal considering both training and validation accuracy, as well as generalizability on test datasets. Although it is early to say if BoW will be adopted by computational ethologists and neuroscientists, our developed framework is also in principle universal to all organisms since it does not rely on specific information of *Hydra*, presenting the stepping stone to developing more sophisticated behavioral methods.

To come clean, as a limitation we should mention that our framework is constrained by the lack of temporal information, which is lost in the bag-of-words approach. Nevertheless, we show that we can still encode *Hydra* behavior even when we do not model the temporal information explicitly. BoW also does not model fine behaviors on the level of single tentacle twitching, or local muscle twitching in body column. This would require an explicit model of the *Hydra* body, instead of the statistical bag-of-words model. Depending on the specific biological question, more specialized method could be developed in the future to investigate these behavior differences.

Revision:

We expanded a paragraph in subsection “A machine learning method for quantifying behavior of deformable animals”, to discuss the general choice of BoW for our behavior recognition task:

“To tackle the problem of measuring behavior in a deformable animal, we developed a novel analysis pipeline using approaches from computer vision that have achieved success in human action classification tasks (Ke et al., 2007; Laptev et al., 2008; Poppe, 2010; Wang et al., 2009, 2011). Such tasks usually involve various actions and observation angles, as well as occlusion and cluttered background. Therefore, they require more robust approaches to capture stationary and motion statistics, compared to using pre-defined template-based features. In particular, the bag-of-words (BoW) framework is an effective approach for extracting visual information from videos of animals with arbitrary motion and deformation. The BoW framework originated from document classification tasks with machine learning. In this framework, documents are considered “bags” of words, and are then represented by a histogram of word counts using a common dictionary. These histogram representations are demonstrated to be efficient for classifying document types. In computer vision, the BoW framework considers pictures or videos as “bags” of visual words such as small patches in the images, or shape and motion features extracted from such patches. Compared with another popular technique, template matching, it is robust against challenges such as occlusion, position, orientation, and viewing angle changes. It also proves to be successful in capturing object features in various scenes and is thus one of the most important concepts and cutting edge techniques in this field. For behavior recognition tasks of deformable animals, it is therefore ideally suited for the problem.”

We modified the following paragraph to discuss the specific modifications we made to the original BoW framework, subsection “A machine learning method for quantifying behavior of deformable animals”:

“We modified the BoW framework by integrating a few state-of-the-art computational methods, including body part segmentation which introduces spatial information, dense trajectory features which encode shape and motion statistics in video patches, Fisher vectors which represent visual words in a statistical manner. Our choice of framework and parameters proves to be quite adequate, considering both its training and validation accuracy, as well as the generalizability on test datasets (Figure 2—figure supplement 1). Indeed, the robust correspondence between supervised, unsupervised and manual classification that we report provides internal cross-validation to the validity and applicability of our machine learning approach. Our developed framework, which uses both supervised and unsupervised techniques, is in principle applicable to all organisms, since it does not rely on specific information of Hydra. Compared with previously developed methods, our method is particularly suitable for behaviors in natural conditions that involve deformable body shapes, as a first step to developing more sophisticated behavioral methods in complex environment for other species.”

We also introduced a paragraph with discussions concerning the potential drawbacks of our method, subsection “A machine learning method for quantifying behavior of deformable animals”:

“Our goal was to describe all possible Hydra behavior quantitatively. Because of this, we chose the bag-of-words framework which captures the overall statistics with a given time frame. We defined the length of basic behavior elements to be 5 seconds, which maximizes the number of behaviors that were kept intact while uncontaminated by other behavior types (Figure 1C–D). The bag-of-words framework has shown success in human action classification tasks; here we improved the basic bag-of-words framework by densely sample feature points in the videos and allowing soft feature quantization with Gaussian Mixture codebook and Fisher vector encoding. However, it should be noted that our approach could not capture fine-level behavior differences, e.g. single tentacle behavior. This would require modeling the animal with an explicit template, or with anatomical landmarks as demonstrated by deformable human body modeling with wearable sensors. Our approach also does not recover transition probabilities between behavior types, or behavioral interactions between individual specimens. In fact, since our method treats each time window as an independent “bag” of visual words, there was no constraint on the temporal smoothness of classified behaviors. Classifications were allowed to be temporally noisy, therefore they could not be applied for temporal structure analysis. A few studies have integrated state-space models for modeling both animal and human behavior (Gallagher et al., 2013; Ogale et al., 2007; Wiltschko et al., 2015), while others have used discriminative models such as Conditional Random Field models for activity recognition (Sminchisescu et al., 2006; Wang and Suter, 2007). These methods may provide promising candidates for modeling behavior with temporal structure in combination with our approach (Poppe, 2010).”

In the Materials and methods section, we added discussions to justify our choice of framework and parameters, as following: subsection “Video pre-processing”:

“… To separate the *Hydra* region into three body parts, the part under the upper body square mask excluding the body column was defined as the tentacle region, and the rest of the mask was split at the minor axis of the ellipse; the part close to the tentacle region was defined as the upper body region, and the other as the lower body region. This step has shown to improve representation efficiency (Figure 2—figure supplement 1B).”

In subsection “Feature extraction”:

“…All parameters above were cross-validated with the training and test datasets.”

And in subsection “Gaussian mixture codebook and Fisher vector”:

“In this paper, the GMM size was set to 128 with cross-validation (Figure 2—figure supplement 1C).”

Along with the revised text, we provided a supplementary figure (Figure 2—figure supplement 1) to justify our specific choices of framework and parameters.

We believe that these modifications together will make our choice of framework and specific steps stronger and will provide a more comprehensive view of our choices.

2) They need to do a better job of motivating and discussing the important questions that their quantitative behavioral repertoire can answer. This is the science of behavior not simply representation of videos as numbers. And it's here that we can learn how Hydra compares to worms and flies what we might expect to find in the neural recordings."

Thank you for the comments. Quantitative behavior recognition and measurement methods provide an important tool for investigating behavioral differences under various conditions from large datasets, allows the discovery of behavior features that are beyond the capability of human visual system, and defines a uniform standard for describing behaviors across conditions. But beyond the purely ethology questions that such methods could answer, they also allow researchers to address potential neural mechanisms by providing a standard and quantitative measurement of the behavioral output of the nervous system. Both ethological and neuroscience application seem important and our approach is quite well poised for these tasks.

Our method also enables the recognition and quantification of *Hydra* behaviors in an automated fashion. Because of this, it provides a quantitative and objective tool to characterize the behavior differences of *Hydra* under pharmacological assays, lesion studies, optogenetic activation of subsets of neurons, or testing the existence of more advanced behaviors such as learning and social behavior. As a proof of concept, it also allows testing quantitative models of behaviors in *Hydra*, investigating the underlying neural activity patterns of each behavior, and predicting the behavioral output from neural activity. As the first pass of its kind, our method opens the possibility to discovery interesting behavioral mechanisms in *Hydra*. And why is Hydra interesting? We would argue that, as a cnidarian, *Hydra’s* nervous system represents one of the earliest nervous systems in evolution. Thus, studying *Hydra* behavior as the output of this primitive nervous system would provide insight into how the nervous system adapts to the changing environment and further evolves.

Revision:

We modified the first and second paragraphs of the Introduction section to integrate the above discussion:

“Animal behaviors are generally characterized by an enormous variability in posture and the motion of different body parts, even if many complex behaviors can be reduced to sequences of simple stereotypical movements (Berman et al., 2014; Branson et al., 2009; Gallagher et al., 2013; Srivastava et al., 2009; Wiltschko et al., 2015; Yamamoto and Koganezawa, 2013). As a way to systematic capture this variability and compositionality, quantitative behavior recognition and measurement methods could provide an important tool for investigating behavioral differences under various conditions from large datasets, allowing for the discovery of behavior features that are beyond the capability of human visual system, and defining a uniform standard for describing behaviors across conditions (Egnor and Branson, 2016). In addition, much remains unknown about how the specific spatiotemporal pattern of activity of the nervous systems integrate external sensory inputs and internal neural network states in order to selectively generate different behavior. Thus automatic methods to measure and classify behavior quantitatively allow researchers to address potential neural mechanisms by providing a standard measurement of the behavioral output of the nervous system.”

“Indeed, advances in calcium imaging techniques have enabled the recording of large neural populations (Chen et al., 2013; Jin et al., 2012; Kralj et al., 2012; St-Pierre et al., 2014; Tian et al., 2009; Yuste and Katz, 1991) and whole brain activity from small organisms such as *C. elegans* and larval zebrafish (Ahrens et al., 2013; Nguyen et al., 2016; Prevedel et al., 2014). A recent study has demonstrated the cnidarian Hydra can be used as an alternative model to image the complete neural activity during behavior (Dupre and Yuste, 2017). As a cnidarian, Hydra is closer to the earliest animals in evolution that possess a nervous system. As an important function of the nervous system, animal behaviors allow individuals to adapt to the environment at a time scale that is much faster than natural selection, and drives rapid evolution of the nervous system, providing a rich context to study nervous system functions and evolution (Anderson and Perona, 2014). As Hydra nervous system evolved from the nervous system present in the last common ancestor of cnidarians and bilaterians, the behaviors of Hydra could also represent some of the most primitive examples of coordination between a nervous system and non-neuronal cells, making it relevant to our understanding of the nervous systems of model organisms such as *C. elegans, Drosophila*, zebrafish, and mice, as it provides an evolutionary perspective to discern whether neural mechanisms found in a particular species represent a specialization or are generally conserved. In fact, although Hydra behavior has been studied for centuries, it is largely unknown whether Hydra possesses complex behaviors such as social behavior and learning behavior, how its behavior changes under environmental, physiological, nutritional or pharmacological manipulations, and the underlying neural mechanisms of the potential changes. Having an unbiased and automated behavior recognition and quantification method would therefore enable such studies with large datasets, allowing us to address the behavioral differences of Hydra with pharmacological assays, lesion studies, environmental and physiological condition changes, under activation of subsets of neurons, testing quantitative models of Hydra behaviors, and linking behavior outputs with the underlying neural activity patterns.”

Reviewer #3 then commented "I think that Reviewer #2 put it well. I would like to see them talk a bit more about:1) The motivations for their representational choices and, importantly, the consequences and implications that these choices have.

If we have to point to a single reason, we chose the current framework specifically because of its advantage of dealing with deformable shapes. But, as discussed in the response to Reviewer #2’s first question, many specific choices in the framework was made due to additional advantages: segmentation and registration eliminate variance caused by background noise and orientation, body part segmentation introduces spatial information to the BoW framework, dense trajectories features maximizes the information captured by the features, Gaussian Mixture codebook and Fisher vectors avoid the inaccuracy of hard encoding from simple k-means codebook and histogram representations. These modifications to the original BoW framework greatly improved the representation efficiency as shown by an overall increase in classification accuracy and generalization ability (Figure 2—figure supplement 1).

As discussed in the response to Reviewer #2’s first question, one important consequence of our framework is the lack of temporal information, which could be done in further work.

Revision:

We made the changes described in answer to Reviewer #2’s first question.

2) How they anticipate using these numbers to answer biological questions. Any measurement representation should have a rational relationship to the questions being asked, so addressing what their method will be useful for (and what it won't be useful for) will be valuable for the literature moving forward. Will this simply be a better detection technique, or does their unsupervised approach allow the field fundamentally new types of behavioral measurements?These concerns and the minor issues from the original review that are appended can form the basis of a revision that clarifies the methodology and brings out the behavioral significance revealed by the new methodology.

Thank you for the comments. Our method could be used to recognize and quantify *Hydra* behaviors from large datasets in an automated and consistent way and would allow us to address questions at the level of behavior repertoire statistics. As discussed in the answer to reviewer #2’s second question, our method provides the possibility to study the behavioral changes of *Hydra* under various conditions such as pharmacological regulations, lesions, environmental and physiological changes. It also provides the tool to investigate the existence of complex behaviors such as social and learning, as well as building and testing quantitative models of *Hydra* behaviors. In this paper, we demonstrated that we can use this method to investigate the behavioral difference under different conditions (e.g. fed/starved). Importantly, our developed framework is not limited to *Hydra*; it is potentially applicable to all animal models since it does not rely on assumptions about the specific features of *Hydra*. However, as pointed out above, our method does lacks temporal information, therefore could not be used to model any particular behavioral sequences and the differences therein.

In addition, the unsupervised approach depends heavily on the quality of encoded features. Since the BoW model provides only a statistical description of videos, the features do not encode fine differences in behaviors. Therefore, the types of behavior that can be identified and quantified by the unsupervised approach have the same constraints as described in the response to Reviewer #2’s first question.

Revision:

Besides the changes we made in response to Reviewer #2’s second question, we added a paragraph in subsection “*Hydra* as a model system for investigating neural circuits underlying behavior”, to address the second part of this comment, concerning the usefulness of our method:

“With our method, we demonstrate that we are able to recognize and quantify Hydra behaviors automatically and identify novel behavior types. This allows us to investigate the behavioral repertoire stability under different environmental, physiological and genetic conditions, providing insight into how a primitive nervous system adapt to its environment. Although our framework does not currently model temporal information directly, it serves as a stepping-stone towards building more comprehensive models of Hydra behaviors. Future work that incorporates temporal models would allow us to quantify behavior sequences, and to potentially investigate more complicated behaviors in Hydra such as social and learning behaviors.”

We also revised the last paragraph of subsection “A machine learning method for quantifying behavior of deformable animals”, in response to the question “will this simply be a better detection technique, or does their unsupervised approach allow the field fundamentally new types of behavioral measurements”:

“In our pipeline, we applied both supervised and unsupervised approaches to characterize Hydra behavior. In supervised classifications (with SVM), we manually defined seven types of behaviors, and trained classifiers to infer the label of unknown samples. In unsupervised analysis (t-SNE), we did not pre-define behavior types, but rather let the algorithm discover the structures that were embedded in the behavior data. In addition, we found that unsupervised learning could discover previously unannotated behavior types such as egestion. However, the types of behaviors discovered by unsupervised analysis are limited by the nature of the encoded feature vectors. Since the bag-of-words model provides only a statistical description of videos, those features do not encode fine differences in behaviors. Due to this difference, we did not apply unsupervised learning to analyze the behavior statistics under different environmental and physiological conditions, as supervised learning appears more suitable for applications where one needs to assign a particular label to a new behavior video.”

[Editors' note: further revisions were requested prior to acceptance, as described below.]

The authors have endeavored to address many of the major concerns of the previous review with further explanations of the method and its underlying assumptions and method choices, and the inherent limitations of the method/analysis. They have also tried to clarify how their method can be used to answer biological questions. There is still one major concern.1a) The authors should try windows greater than 5 seconds. It's hardly surprising that less than five seconds is less effective, but why not 8, 10, 20? Just saying "we noticed 5 seconds is a reasonable length to define a single behavior" is hardly convincing (neither is Figure 1C).

We now tested window size of 8 seconds, 10 seconds and 20 seconds with our developed analysis framework, and compared the training, validation and test classification accuracy. The result shows that 5-second time window still performs best with all the accuracy measurements (Figure2—figure supplement 1A). Therefore, we believe 5-second is a reasonable length to define a single behavior with our method.

Revision:

We modified Figure2—figure supplement 1A to include the classification accuracy of window sizes greater than 5 seconds, and modified the corresponding text (subsection “Capturing the movement and shape statistics of freely-moving *Hydra”*):

“Our goal was to […] A post hoc comparison of different window sizes (1-20 seconds) with the complete analysis framework also demonstrated that 5-second windows result in the best performance (Figure 2—figure supplement 1A). Therefore, we chose 5-second as the length of a behavior element in Hydra.”

We also modified the corresponding Figure legend.

1b) It still remains possible that the highly-fragmented t-SNE representation results form the fact that behaviors are unnecessarily chopped-up by imposing a 5 second window. Problems might occur because the behavior spans a window boundary. The analysis should be performed using a sliding 5-second window rather than separated windows. This may remove some of the observed over-segmentation of the space. There are several methods (including one in Berman, 2014, but others as well) for handling tens to hundreds of millions of data points. Since the space is one of the cruxes of the paper's arguments, and the authors might get better results with the sliding window, it seems somewhat remiss to not attempt this (it would be ~ 24 hours of running time on a machine that can handle a 30,000-point t-SNE). The Barnes-Hut implementation from: https://lvdmaaten.github.io/tsne/ may prove helpful.

We performed t-SNE with the fast Barnes-Hut implementation on the complete training dataset of 50 *Hydra*, with a sliding 5-second window. The resulting space did not show improved segmentation (see Author response image 1). It is possible that, by performing sliding windows, the highly-overlapping windows introduce more local structures that represent the similarities within each individual, rather than within each behavior category. As t-SNE is designed for discovering local similarity structures, this results in the highly segmented embedding map as shown below. We believe that the original embedding analysis presented in the manuscript represents a proof-of-concept demonstration of categorizing behavior types with unsupervised methods, without too much bias created by individual similarities.

**Author response image 1. respfig1:** t-SNE embedding of continuous time windows. a, Scatter plot with embedded Fisher vectors from 50 *Hydra*. Each dot represents projection from a high-dimensional Fisher vector to its equivalent in the embedding space. The Fisher vectors were encoded from continuous 5-second windows with an overlap of 24 frames. Color represents the manual label of each dot. b, Segmented density map generated from the embedding scatter plot. c, Behavior motif regions defined using the segmented density map. d, Labeled behavior regions with manual labels. Color represents the corresponding behavior type of each region.